# Diffusion Guided Adversarial State Perturbations in Reinforcement Learning

**Xiaolin Sun[1], Feidi Liu[2], Zhengming Ding[1], and Zizhan Zheng[1]**

[1]Department of Computer Science, Tulane University
[2]Shanghai Center for Mathematical Science, Fudan University
{xsun12,zding1,zzheng3}@tulane.edu, fdliu23@m.fudan.edu.cn

## Abstract

Reinforcement learning (RL) systems, while achieving remarkable success across various domains, are vulnerable to adversarial attacks. This is especially a concern in vision-based environments where minor manipulations of high-dimensional image inputs can easily mislead the agent's behavior. To this end, various defenses have been proposed recently, with state-of-the-art approaches achieving robust performance even under large state perturbations. However, after closer investigation, we found that the effectiveness of the current defenses is due to a fundamental weakness of the existing $l_p$ norm-constrained attacks, which can barely alter the semantics of image input even under a relatively large perturbation budget. In this work, we propose SHIFT, a novel policy-agnostic diffusion-based state perturbation attack to go beyond this limitation. Our attack is able to generate perturbed states that are semantically different from the true states while remaining realistic and history-aligned to avoid detection. Evaluations show that our attack effectively breaks existing defenses, including the most sophisticated ones, significantly outperforming existing attacks while being more perceptually stealthy. The results highlight the vulnerability of RL agents to semantics-aware adversarial perturbations, indicating the importance of developing more robust policies. Our code can be found at this `GitHub Repo`.

## 1 Introduction

Reinforcement learning (RL) has seen significant advancements in recent years, becoming a key area of machine learning. RL's ability to enable agents to learn optimal decision-making policies through interaction with dynamic environments have led to breakthroughs in various fields. Beginning from AlphaGo [44], RL-based systems show the ability to surpass human performance in complex games. Beyond gaming, RL is driving innovations in robotics, self-driving cars [30], and industrial automation, where agents learn to navigate, manipulate, and interact autonomously.

However, RL is vulnerable to various types of attacks, such as reward and state perturbations, action space manipulations, and model inference and poisoning [26]. Recent studies have shown that an RL agent can be manipulated by perturbing its observation [24, 60] and reward signals [25], and a well-trained RL agent can be confounded by a malicious opponent behaving unexpectedly [17]. In particular, a malicious agent can subtly manipulate the observations of a trained RL agent, resulting in a significant drop in performance and cumulative reward [60, 47]. Such attacks exploit vulnerabilities in the agent's perception systems, including sensors and communication channels, without needing to cause obvious disruptions. This susceptibility to minor perturbations raises major concerns, particularly for RL applications in security-sensitive and safety-critical environments.

39th Conference on Neural Information Processing Systems (NeurIPS 2025).

Multiple defenses have been proposed to mitigate state perturbation attacks. Regularization-based methods like SA-MDP [60] and WocaR-MDP [36] improve robustness by smoothing policies and estimating worst-case rewards. CAR-DQN [35] further enhances robustness using the Bellman Infinity-error. Adversarial training methods including alternating training [61] and game-theoretical methods like PROTECTED [39] and GRAD [37] can potentially lead to more robust policies but are costly and do not scale to image-based inputs. More recently, diffusion-based methods, such as DMBP [57] and DP-DQN [46], improve robustness by reconstructing the true states or modeling beliefs about them through denoising. These advanced defenses can withstand state-of-the-art attacks like PGD [60], MinBest [24], PA-AD [47], and high-sensitivity direction attacks [31].

However, we found that current attacks share two major shortcomings when applied to agents with raw pixel images as input such as autonomous driving [30] and embodied AI [58]. First, with the exception of Korkmaz [31], current attacks usually restrict a perturbed state to be within an $\epsilon$-ball of the true state, measured using an $l_p$ norm, to avoid detection. However, this approach struggles with producing *realistic* perturbed input that is within the natural data distribution and further constrains the attacker's search space for generating semantics-changing perturbations. While this problem has been considered in supervised settings [54, 15, 51, 13], to the best of our knowledge, Korkmaz [31] was the only prior study that identified this limitation for state perturbations in RL. However, the high-sensitivity direction attacks proposed in Korkmaz [31] adopt simple image transformations that mainly target changes in visually significant but not domain-specific semantics. Consequently, the perturbed states either can be denoised by diffusion-based defenses or are non-stealthy from human perspectives (see our evaluation results in Appendix D). Second, previous attacks mainly focus on improving attack performance while ignoring the temporal dependencies across states, which is unique to RL. As a result, they often generate states that are inconsistent with the agent's previous observations (either perturbed or not). Recently, Liang et al. [37] looked into this problem by considering two consecutive time steps, but it still utilized an $l_p$-norm constraint. On the other hand, Illusory attacks [16] require a perturbed trajectory to follow the same distribution as the normal trajectory, making it difficult to detect. However, this approach does not scale to high-dimensional image input. None of these attacks can easily modify the domain-specific semantics of the image input while keeping it realistic and plausible. The perturbed states generated by these attacks can be easily denoised with the help of a history-conditioned diffusion model.

With these two shortcomings in mind, we propose SHIFT (**S**tealthy **H**istory-al**I**gned di**F**fusion a**T**tack), a novel semantics-aware and policy-agnostic attack method that goes beyond the traditional $l_p$ norm constraint. Our approach is grounded in precise definitions of realistic states and three attack properties: semantic-altering, historically-aligned, and trajectory-faithful, which provide novel characterizations of semantics-aware and stealthy attacks in sequential decision-making. As these metrics are computationally expensive to evaluate, we provide practical methods to approximate them. Our main contribution is the development of a diffusion-based attack

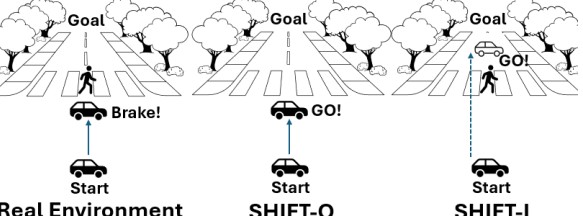

Figure 1: A car approaches a crosswalk with a pedestrian ahead. The safe, optimal action is to brake. **SHIFT-O** removes the pedestrian from the agent's observation, while **SHIFT-I** creates an imaginary trajectory suggesting the car has already crossed. Both mislead the agent into moving forward, resulting in a collision.

framework that utilizes classifier-free guidance to approximate history-aligned state generation, which is further improved using policy guidance to generate effective, realistic, and stealthy state perturbations.

Using this novel guided diffusion approach, we propose two versions of **SHIFT**: **SHIFT-O** perturbs the image input conditioned on the actual history to immediately induce suboptimal actions, while **SHIFT-I** guides the agent toward an imagined trajectory that is self-consistent, but ultimately leads to poor performance when actions are executed in the real environment. We compare these two methods in Figure 1, with a working example in the Freeway environment given in Figure 6 in Appendix C. SHIFT is policy-agnostic, capable of adapting to multiple victim policies without retraining, and applicable to both value-based and policy-based reinforcement learning methods.

Comprehensive evaluations show that it can break all known defenses, lower agents' cumulative reward in various environments by more than 50%, while being stealthier than prior attacks, as shown by lower reconstruction error, Wasserstein-1 distance, and LPIPS [62], and higher SSIM [50]. Our results highlight that RL agents with image input are vulnerable to semantics-aware adversarial perturbations, which has important implications when deploying them in sensitive domains.

## 2 Preliminary

### 2.1 Reinforcement Learning (RL) and State Perturbation Attacks

An RL environment can be formulated as a Markov Decision Process (MDP), denoted as a tuple $\langle S, A, P, R, \gamma, \rho_0 \rangle$, where $S$ is the state space and $A$ is the action space. $P : S \times A \to \Delta(S)$ is the transition function, where $P(s'|s, a)$ denotes the probability of moving to state $s'$ given the current state $s$ and action $a$. $R : S \times A \to \mathbb{R}$ is the reward function where $R(s, a) = \mathbb{E}(R_t|s_{t-1} = s, a_{t-1} = a)$ and $R_t$ is the reward in time step $t$. Finally, $\gamma$ is the discount factor and $\rho_0$ is the initial state distribution. An RL agent wants to maximize its cumulative reward $G = \Sigma_{t=0}^{T} \gamma^t R_t$ over a time horizon $T \in \mathbb{Z}^+ \cup \{\infty\}$, by finding a (stationary) policy $\pi : S \to \Delta(A)$.

First introduced in [24], a **state perturbation attack** is a test-stage attack targeting an RL agent with a well-trained policy $\pi$. We consider the worst-case scenario where the attacker has access to a clean environment and the victim's policy $\pi$ and all deployed defense mechanisms. Further, the attacker has access to the true states in real-time. At each time step, the attacker observes the true state $s_t$ and generates a perturbed state $\tilde{s}_t$. The agent, however, only observes $\tilde{s}_t$ (and not $s_t$) and takes an action $a_t$ based on its policy $\pi(\cdot|\tilde{s}_t)$. Note that the attacker only interferes with the agent's observed state and does not modify the underlying MDP. Consequently, the true state at the next time step is governed by the transition dynamics $P(s_{t+1}|s_t, \pi(\cdot|\tilde{s}_t))$. The attacker's objective is to minimize the agent's long-term cumulative reward. Let $o_t$ denote the observation to the agent, which can be either $s_t$ or $\tilde{s}_t$, depending on whether there is an attack at time $t$.

Further, the attacker needs to remain stealthy to avoid immediate detection and achieve its long-term goal. To this end, previous state perturbation attacks [60, 47] restrict the attacker's ability by a budget $\epsilon$, so that $\tilde{s}_t \in B_\epsilon(s_t)$ where $B_\epsilon(s_t)$ is the $l_p$ ball centered at $s_t$ for some norm $p$ (typically an $l_\infty$ norm is used). However, state-of-the-art diffusion-based defenses [46] are able to mitigate the restricted attack even with a large $\epsilon$. Further, existing efforts [31] that try to go beyond the $l_p$-norm constraint cannot compromise these advanced defenses without being detected. A detailed discussion on related work is in Appendix A.

### 2.2 Denoising Diffusion Probabilistic Model (DDPM)

Diffusion models, particularly Denoising Diffusion Probabilistic Models (DDPMs), have recently gained attention as generative models that iteratively reverse a predefined diffusion process to generate data from noise [22]. A DDPM model consists of two phases: a forward diffusion process that gradually adds noise to the data and a reverse denoising process to recover the original data.

**Forward Process.** The forward process is a fixed Markov chain that progressively corrupts the data $\mathbf{x}_0$ over $T$ time steps by adding Gaussian noise. At each step, the data evolve according to $q(x_i \mid x_{i-1}) = \mathcal{N}\left(x_i; \sqrt{1 - \beta_i} x_{i-1}, \beta_i \mathbf{I}\right)$, where $\beta_i \in (0, 1)$ controls the noise level at step $i$.

**Reverse Process.** The reverse process manages to recover the data $x_0$ from the noisy sample $x_T$. The reverse process is another Markov chain, parameterized by a neural network $\epsilon_\theta(x_i, i)$, which predicts the noise added to the data at each time step $i$ in the forward process and can be modeled as:

$$p_\theta(x_{i-1} \mid x_i) = \mathcal{N}\left(x_{i-1}; \mu_\theta(x_i, i), \sigma_\theta^2(x_i, i)\mathbf{I}\right), \tag{1}$$

where $\mu_\theta$ is the predicted mean and $\sigma_\theta^2$ is the variance of the reverse distribution at each time step $i$.

**Training.** The training goal of DDPM is to learn a model $\epsilon_\theta(x_i, i)$ that predicts the noise added to a data point $x_0$ during the forward diffusion process. $\mu_\theta(x_i, i)$ in the reverse process is expressed in terms of the predicted noise $\epsilon_\theta(x_i, i)$:

$$\mu_\theta(x_i, i) = \frac{1}{\sqrt{1 - \beta_i}} \left( x_i - \frac{\beta_i}{\sqrt{1 - \bar{\alpha}_i}} \epsilon_\theta(x_i, i) \right), \tag{2}$$

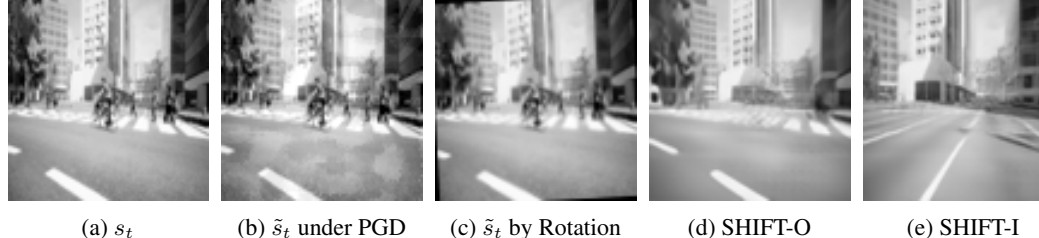

| (a) $s_t$ | (b) $\tilde{s}_t$ under PGD | (c) $\tilde{s}_t$ by Rotation | (d) SHIFT-O | (e) SHIFT-I |

Figure 2: Examples of true and perturbed states captured by the front camera of a vehicle in the AirSim driving simulator. a) is the true state. b) and c) are the perturbed states under the PGD attack with a $l_\infty$ budget of $\frac{15}{255}$ and through rotation [31] by 3 degrees counterclockwise, respectively. d) and e) are the perturbed states generated by our SHIFT-O and SHIFT-I attacks, respectively. Neither PGD nor Rotation attacks can alter the decision-related semantics. SHIFT-O removed pedestrians and bicycles at the crosswalk while being aligned with the real history. SHIFT-I lures the driver into thinking that the car has already crossed the crosswalk, when in fact it has not in the real environment. Note that SHIFT-I is history-aligned with the observed trajectory but not the real trajectory.

where $\bar{\alpha}_i = \prod_{n=1}^{i}(1 - \beta_n)$ is the cumulative product of $(1 - \beta_i)$ over time steps.

The variance $\sigma_\theta^2(x_i, i)$ can be predicted through a neural network or set by a predetermined scheduler. In DDPM, $\sigma_\theta^2(x_i, i)$ is set according to a fixed schedule as $\sigma_i^2 = \beta_i$. The DDPM's training loss can be written as $\mathcal{L}_{\text{simple}} = \mathbb{E}_{x_0, i, \epsilon} \left[ \|\epsilon - \epsilon_\theta(x_i, i)\|^2 \right]$, where $\epsilon \sim \mathcal{N}(0, \mathbf{I})$ is the noise added during the forward process. By minimizing this loss, the model learns to iteratively remove noise from $x_i$, ultimately generating high-quality samples from the learned data distribution.

## 3 Stealthy History-Aligned Diffusion Attack

We introduce SHIFT, a novel policy-agnostic state perturbation attack built upon diffusion models that combines the classifier-free and policy guidance methods. We first discuss the motivation for using diffusion models to generate perturbed states, where we also formulate the attack objectives by giving a novel characterization of realistic perturbed states and three attack properties: semantic-altering, historically-aligned, and trajectory-faithful (Section 3.1). We then discuss how to achieve these properties and remain realistic in Section 3.2.

### 3.1 Motivations and Attack Objectives

State-of-the-art perturbation attacks against image input (such as PGD, MinBest, and PA-AD) are performed by adding $l_p$-norm constrained noise. While the high-sensitivity direction attacks [31] can go beyond the $l_p$-norm constraint, they are implemented through simple image transformations such as rotations and shifts that often change the output layout (see Figure 2c and Figure 7 in the Appendix). In both cases, the perturbed states often fall outside the set of states that can be generated by the underlying MDP (determined by the environment physics engine). Consider the front camera snapshots from an autonomous driving agent in Airsim [43] simulator shown in Figure 2. Figure 2b shows that a perturbed state generated by the PGD attack with an $l_\infty$ budget of $\epsilon = \frac{15}{255}$ is easily recognizable due to noticeable noise, making the observation appear unnatural and unrealistic. On the other hand, smaller perturbations are ineffective, especially in the presence of strong defenses. The main reason is that these attacks typically cannot induce semantic changes. As shown in Figure 2b, even with a relatively large attack budget ($\epsilon = \frac{15}{255}$), the pedestrians maintain their positions.

Our objective is to go beyond $l_p$ norm-constrained attacks to generate more powerful and stealthy state perturbations. A key observation is that a carefully designed diffusion model can enable more effective attacks by generating semantics-changing state perturbations (e.g., Figure 2) to mislead the victim to choose a suboptimal action, leading to significant performance loss. To generate realistic perturbed states that are semantically different from original states, we introduce the following definitions.

**Definition 3.1 (Valid States)** *The set of valid states $S^*$ of an MDP $\langle S, A, P, R, \gamma, \rho_0 \rangle$ is defined as: $S^* \coloneqq \{s \in S \mid \exists \pi \in \Pi, d_\pi(s) > 0\}$, where $\Pi$ denotes the set of all possible (stationary) policies and $d_\pi(s)$ represents the stationary state distribution under policy $\pi$, from the initial distribution $\rho_0$.*

In other words, $S^*$ consists of all states that can be reached by following an arbitrary policy $\pi$ from the initial distribution $\rho_0$. However, ensuring strict validity is intractable with limited amount of data (as in the case of diffusion models). Thus, we introduce the concept of **realistic states** as a more practical measure, based on the projection distance between a state $s$ and the set of valid states $S^*$.

**Definition 3.2 (Realistic States)** *A state $s$ is defined as realistic if its projection distance to the set of valid states $S^*$ is bounded by a threshold $\delta$. Formally, the set of realistic states $S^r$ is defined as: $S^r := \{s \in S \mid \|Proj_{S^*}(s) - s\|_2 \leq \delta_1\}$, where $Proj_{S^*}(s) = \arg\min_{s' \in S^*} \|s' - s\|_2$ is the projection of $s$ onto $S^*$, and $\delta_1$ is a predefined threshold.*

Similar to the $l_p$ norm constraint, the realistic states constraint prevents the attack from generating arbitrary, meaningless perturbations, while still allowing semantically meaningful state changes. While the above condition captures our intuition on realism, it is computationally expensive to verify since the set of valid states $S^*$ is typically unknown and hard to estimate. Thus, we approximate the projection distance using the reconstruction error from an autoencoder-based anomaly detector, which will be discussed in Section 3.2.3. We consider realistic states to be indistinguishable from valid states and pose realism as an objective of our approach. With the above definition, we further define a perturbed state $\tilde{s}$ to be semantically different from the original state $s$ if its projection to $S^*$ differs from $s$. Let $D(\tilde{s})$ denote the set of projection points of $\tilde{s}$ onto $S^*$, which might not be unique depending on the state space $S^*$. The condition can be formally stated as follows.

**Definition 3.3 (Semantics-Changing States)** *A perturbed state $\tilde{s}$ is semantically different from the true state $s$ when $\exists \tilde{s}' \in D(\tilde{s}), \tilde{s}' \neq s$.*

While the above definitions capture the quality of individual states, they ignore the dynamic nature of sequential decision-making in RL and may lead to perturbed states that significantly deviate from history. Thus, we look for perturbed states that not only change the semantic meaning but also align with a given history, formally defined as follows.

**Definition 3.4 (History-Aligned States)** *Let $H_{t-1} := (o_{t-k}, \ldots, o_{t-1})$ denote the sequence of last $k$ observed states by the agent (after projection onto $S^*$). Given the agent's policy $\pi$, a perturbed state $\tilde{s}_t$ at time step $t$ is aligned with $H_{t-1}$ from the agent's view if:*

$$\tilde{s}_t \in S(H_{t-1}) := \{\tilde{s}_t \in S \mid \Pr_\pi(S_t \in D(\tilde{s}_t) \mid H_{t-1}) > 0\},$$

*where $S_t$ is the random variable for the true state at $t$.*

That is, $\tilde{s}_t$ is aligned with the history if any projection point in $D(\tilde{s}_t)$ is a reachable next state given the victim's observed projected history. This ensures that the perturbed state remains undetectable even when the agent employs a history-based detector. Note that instead of using the $\text{Proj}_{S^*}(\cdot)$ operator, the above definition can be extended to incorporate the actual detector (if there is any) used by the agent. Similar to our realism measure, the above condition can be challenging to directly verify and optimize. We extend the concept of Wasserstein adversarial examples [51] to the RL setting to approximately measure history alignment in our experiments as discussed in Section 4. We note that perturbations that are perfectly aligned with history are hard to detect without access to true states. On the other hand, a history-aligned state may deviate significantly from the true state, as shown in Figure 2e, and it is not stealthy when the agent does have some knowledge about the true state. To capture the latter case, we introduce the concept of trajectory faithfulness as defined below.

**Definition 3.5 (Trajectory Faithfulness)** *Let $H_{t-1} := (o_{t-k}, \ldots, o_{t-1})$ denote the sequence of the last $k$ observed states by the agent (after projection onto $S^*$). The observed states are **trajectory faithful** if $\sum_{i=t-k}^{t-1} \|o_i - s_i\|_2 \leq \delta_2$, where $s_i$ denotes the true state, and $\delta_2$ is a faithfulness threshold.*

Figure 2d slightly alters the semantics of the true state by removing the pedestrians while keeping other elements consistent with the true state, improving faithfulness compared to Figure 2e. Note that it is hard to directly calculate trajectory faithfulness due to the projection. Instead, we use SSIM [50] and LPIPS [62] between the perturbed states and true states as quantified metrics.

## 3.2 Diffusion-based State Perturbations

In this section, we discuss SHIFT, our diffusion-based attack that can generate realistic and stealthy state perturbations with three properties defined above: Semantic Change, Historical Alignment,

and Trajectory Faithfulness. Our attack consists of two stages: the training stage and the testing stage. During the training stage, we use data generated by the clean environment (i.e., the MDP) to train a conditional diffusion model to generate states that are realistic and history-aligned. We further train an autoencoder to detect unrealistic states. In the testing stage, we employ the pretrained diffusion model to generate perturbed states guided by (1) the defender's policy $\pi$ and corresponding state-action value function $Q^\pi$, which provides guidance toward a perturbed state that has lower $Q^\pi(s, \pi(\tilde{s}))$ value, and (2) the pre-trained autoencoder, which further enhances the realism of the perturbed states, (3) a given history to be aligned with. Figure 4 in the appendix illustrates the two stages of our attack and the main components involved, with each discussed in detail below.

### 3.2.1 Generating History-Aligned States via Conditional Diffusion

We first describe the training of the conditional diffusion model, which is built upon the classifier-free guidance approach [21] that can generate both unconditional and conditional samples, enabling the model to guide itself during the generation process. We train a classifier-free guidance model conditioned on a history to generate the next state $\tilde{s}_t$ that follows the given history. This ensures that the generated next state $\tilde{s}_t$ is realistic and aligned with the history, such that $\tilde{s}_t$ is stealthy from both the static and dynamic views. It is important to note that the true next state $s_t$ is independent of the victim's policy $\pi$ when the history (including previous states and actions) is given. Thus, we can train this diffusion model with classifier-free guidance without requiring knowledge of the specific victim's policy $\pi$. However, a separate diffusion model needs to be trained for each distinct environment.

Specifically, let $\tau_{t-1} = \{s_{t-k}, a_{t-k}, ..., s_{t-1}, a_{t-1}\}$ be the **true** history from time $t - k$ to time $t$, where $k > 1$ is a parameter. In our setting, the model is trained with both class-conditional data $(s_t, \tau_{t-1})$ and unconditional data $s_t$ by randomly dropping $\tau_{t-1}$ with a certain probability. The history $\tau_{t-1}$ and true state $s_t$ are sampled from trajectories generated in a clean environment by following a well-trained policy $\pi_{ref}$ (independent of the agent's policy) with exploration to ensure coverage. The noise prediction network $\epsilon_\theta(s_t^i, i, \tau_{t-1})$ is trained to learn both conditional and unconditional distributions during the training. When generating perturbed states at the testing stage, where the reverse process is applied, the noise prediction can be adjusted using a guidance scale $\Gamma(i)$ as follows:

$$\epsilon_i = \Gamma(i)\epsilon_\theta(s_t^i, i, \tau_{t-1}) + (1 - \Gamma(i))\epsilon_\theta(s_t^i, i), \tag{3}$$

where $\tau_{t-1}$ is the given history, and $\Gamma(i)$ controls the strength of the guidance. Note that we have two time step variables here, where $t$ is the time step in an RL episode and $i$ is the index of the reverse steps in the reverse process. With classifier-free guidance, the model learns a distribution of $\tilde{s}_t$ conditioned on a historical trajectory $\tau_{t-1}$. The attacker can set a given history $\tau_{t-1}$ as a conditioning factor, forcing the generated perturbed state $\tilde{s}_t$ to align with $\tau_{t-1}$. Consequently, the classifier-free guidance enhances dynamic stealthiness and achieve historical alignment.

Since the classifier-free model is designed to generate the true next state $s_t$ based on the history $\tau_{t-1}$, the generated next state $\tilde{s}_t$ is expected to be close to the true state $s_t$. Consequently, while the generated next state $\tilde{s}_t$ may not be exactly the same as $s_t$ to be classified as a valid state, $\tilde{s}_t$ is sufficiently close to $s_t$ to be considered as a realistic state according to Definition 3.2. This is confirmed in Figure 3c, which shows the average $l_2$ distance between the perturbed states generated through the conditional diffusion model and the true states in the Freeway environment. Note that the $l_2$ distance gives an upper bound on the realism measure in Definition 3.2 as the true state may not be the closest state in $S^*$ with respect to the perturbed state. The results show that states generated by the diffusion model conditioned on history are closer to the true states compared to those generated by PGD, even with small budgets, which highlights the realism of our method.

### 3.2.2 Generating Semantics Changing Perturbations via Policy Guidance

A perturbed state $\tilde{s}_t$ that is solely generated by the history-conditioned guidance can not mislead the victim toward a suboptimal action. To achieve our attack objective of decreasing agent performance, we introduce a policy guidance module that is similar to classifier guidance at the testing stage that can change the semantics of the true state $s_t$. Classifier guidance is a method to improve the quality of samples generated by incorporating class-conditional information [14]. The core idea is to utilize a pre-trained classifier $p_\Phi(y|\mathbf{x})$, where $y$ represents the class label, to guide the reverse diffusion process toward generating samples conditioned on a desired class. In our context, we can use the state-action value function of the victim's policy $Q^\pi(s, a)$ for guidance. Since the policy guidance is

applied during testing and is independent of the pre-trained conditional diffusion model, our approach remains policy-agnostic, allowing it to effectively adapt to multiple victim policies without retraining.

At each reverse time step $i$, the reverse process is modified by adjusting the mean of the noise prediction model $\epsilon_i$ with the gradient of the policy with respect to $\tilde{s}_t^i$, that is, $-\nabla_{\tilde{s}_t^i} \log Q^\pi(s_t, \pi(\tilde{s}_t^i))$. This guidance steers the generation process towards samples that lead the agent take actions that have lower state-action value under true states, ultimately changing the semantics and causing victim's performance drop at the same time. As shown in [14], for the unconditional reverse transition $p_\theta$ in (1), the modified reverse process with policy guidance can be expressed as: $p(\tilde{s}_t^{i-1} \mid \tilde{s}_t^i, Q^\pi) = \mathcal{N}\left(\tilde{s}_t^{i-1}; \mu_\theta(\tilde{s}_t^i, i) - \sigma_i^2 \nabla_{\tilde{s}_t^i} \log Q^\pi(s_t, \pi(\tilde{s}_t^i)), \sigma_i^2 \mathbf{I}\right)$. In our scenario, however, policy guidance is applied to a diffusion model conditioned on the history $\tau_{t-1}$. Typically, policy guidance cannot be applied to a conditional diffusion model because the gradient term becomes $-\nabla_{\tilde{s}_t^i} \log Q^\pi(s_t, \pi(\tilde{s}_t^i)|\tau_{t-1})$, which is hard to compute through $Q^\pi$. Fortunately, in our RL setting, policy guidance and classifier-free guidance can be combined as shown in the following theorem.

**Theorem 3.6** *The reverse process when sampling from a history-conditioned DDPM model guided by the victim's state-action value function $Q^\pi$ is given by*

$$p(\tilde{s}_t^{i-1} \mid \tilde{s}_t^i, Q^\pi, \tau_{t-1}) = \mathcal{N}\left(\tilde{s}_t^{i-1}; \mu_i - \sigma_i^2 \nabla_{\tilde{s}_t^i} \log Q^\pi(s_t, \pi(\tilde{s}_t^i)), \sigma_i^2 \mathbf{I}\right),$$

*where $\mu_i$ is derived from $\epsilon_i$ in (3), as given by (2), and $\sigma_i^2$ is determined by the variance scheduler $\beta_i$.*

Theorem 3.6 shows that policy guidance and classifier-free methods can coexist without interference. While this is generally not true, it holds in our setting because given the two conditioning variables $Q^\pi$ and $\tau_{t-1}$, the noise predicted by classifier-free guidance depends only on $\tau_{t-1}$, while the gradient from policy guidance depends solely on $Q^\pi$ (Proof in Appendix B). With policy guidance modifying the reverse process, the perturbed state $\tilde{s}_t$, conditioned on $(Q^\pi, \tau_{t-1})$, differs from states conditioned only on $\tau_{t-1}$. Thus, $\tilde{s}_t$ is semantically distinct from the true state $s_t$ and achieves attack performance.

### 3.2.3 Enhancing Realism via Autoencoder Guidance

Since the classifier guidance method introduces additional gradient information during the reverse process, the generated perturbed state $\tilde{s}_t$ may deviate from realistic states. To address this, we incorporate an autoencoder-based anomaly detector trained on clean data. Autoencoders [66] are widely used in unsupervised anomaly detection by measuring reconstruction error—defined as the $l_2$ distance between an input state $s_t$ and its reconstruction, which is $\mathcal{L}(s_t, \mathbf{AE}(s_t)) = \|s_t - \mathbf{AE}(s_t)\|_2$. Since the autoencoder $\mathbf{AE}(\cdot)$ is trained solely on clean states, it assigns significantly higher errors to anomalous or unrealistic inputs, providing an effective signal to assess the realism of generated states. Thus, we can use the pre-trained autoencoder at the testing stage to enhance the realism of the perturbed states, following policy guidance, which can be achieved by $\tilde{s}_t^i = \tilde{s}_t^i - \nabla_{\tilde{s}_t^i} \mathcal{L}(\tilde{s}_t^i, \mathbf{AE}(\tilde{s}_t^i))$.

### 3.2.4 SHIFT-O/I–Tradeoff between Historical Alignment and Trajectory Faithfulness

Both **SHIFT-O** and **SHIFT-I** share the same pre-trained diffusion model and the policy and realism guidance modules, differing only in their history conditioning during attacks. **SHIFT-O** uses the true history $\tau_{t-1} = \{s_{t-k}, a_{t-k}, \ldots, s_{t-1}, a_{t-1}\}$ as the condition to generate a perturbed state that remains aligned with the true trajectory. In contrast, **SHIFT-I** uses the victim's observed history (i.e., the perturbed history), $\tilde{\tau}_{t-1} = \{o_{t-k}, a_{t-k}, \ldots, o_{t-1}, \varnothing\}$, with the last action dropped, as the condition. Thus, **SHIFT-I** generates perturbed states that are aligned with the observed trajectory. Dropping the last action grants the model greater flexibility to sample perturbed states that might follow alternative, suboptimal actions, thereby increasing the likelihood of misleading the agent.

**SHIFT-I** aligns with the (perturbed) history and can alter decision-relevant semantics by conditioning on the perturbed history. It maintains self-consistent, policy-guided imaginary trajectories, making the attack difficult to detect when the agent lacks access to the true states. This represents a realistic attack scenario, when attacks occur over a short time span and harmful consequences may already have unfolded before the agent realizes it was misled in the last few time steps. A similar illusory attack was recently proposed in [16]; however, it does not scale to image input. Further, such attacks can lead to trajectories that deviate significantly from the true ones, as shown in Figure 2, and are therefore detectable when the agent has access to even a single true state before the attack succeeds.

**SHIFT-O** improves trajectory-faithfulness by conditioning on true historical states, while altering semantics via policy guidance (e.g., removing pedestrians v.s. shifting the whole scene). However, these semantic changes may introduce subtle historical inconsistencies as abrupt object disappearance will disrupt temporal coherence, resulting in imperfect historical alignment. Such inconsistency can potentially be detected by an agent with sufficient domain knowledge and advanced detection capabilities. However, as we show in our evaluations, this is a nontrivial task for complex environments.

Traditional $l_p$-norm constrained attacks (e.g., PGD, MinBest, PA-AD) are history aligned and trajectory faithful under a small attack budget, by introducing minimal pixel-level perturbations. Similarly, high-sensitivity direction attacks from Korkmaz [31] maintain these two properties by applying simple image transformations that alter visually significant but not domain-specific semantics. Their lack of semantic-altering capability causes them to be ineffective against diffusion-based defenses.

Our framework reveals a fundamental *trilemma* in adversarial perturbations in sequential decision-making with image input. Current methods are optimized for only two out of three critical properties: semantic-altering (Def 3.3), historically-aligned (Def 3.4), and trajectory-faithful (Def 3.5). Finding an attack that better satisfies all three properties than current methods remains an open problem.

### 3.3   Implementation Details

Previous attack methods, such as PGD and PA-AD, assume the attack happens at every time step. In contrast, **SHIFT** is only allowed to attack a fraction $\xi$ of the time steps. To determine when to attack, we adopt the concept of *state importance weight* from Liang et al. [36], defined as $\omega(s) = \max_{a_1 \in A} Q(s, a_1) - \min_{a_2 \in A} Q(s, a_2)$. At time step $t$, SHIFT computes $\omega(s_t)$ and compares it to the top $\xi$-th percentile of all previous importance weights. If $\omega(s_t)$ falls within the top $\xi$ percentile and the number of attacks so far is less than $t \cdot \xi$, SHIFT injects an attack at step $t$. It is worth noting that **SHIFT-O** passes the true state to the agent when no attack is injected. In contrast, **SHIFT-I** continues to generate states conditioned on the perturbed history and true actions without policy guidance, in order to maintain consistency within the imaginary trajectory.

Although our theoretical analysis in Section 3.2.2 uses the DDPM method for simplicity, we implement our attack using the EDM formulation [29] which is a score-based diffusion method that requires fewer reverse diffusion steps, improving sampling efficiency and enabling real-time attacks. We build upon previous work [3], which applied EDM-based conditional diffusion models to Atari world modeling. We apply the technique from [5] to optimize policy guidance, which first calculates the output sample $\hat{s}_t$ without any attack, then apply policy guidance to guide the reverse process. Details on the EDM formulation and implementation are in Appendix C. We provide algorithms for training and sampling of our attack in Algorithms 1 and 2 in Appendix C.6.

Table 1: Episode reward of **SHIFT-O** and **SHIFT-I** against various defense methods in different environments with attack frequency 1.0 and 0.25. All results are reported with mean and std over 10 runs.

| | Pong | | | | Freeway | | | |
|---|---|---|---|---|---|---|---|---|
| **Model** | **SHIFT-O-1.0** | **SHIFT-O-0.25** | **SHIFT-I-1.0** | **SHIFT-I-0.25** | **SHIFT-O-1.0** | **SHIFT-O-0.25** | **SHIFT-I-1.0** | **SHIFT-I-0.25** |
| **DQN-No Attack** | 21.0±0.0 | 21.0±0.0 | 21.0±0.0 | 21.0±0.0 | 34.1±0.1 | 34.1±0.1 | 34.1±0.1 | 34.1±0.1 |
| **DQN** | -21.0±0.0 | -20.8±0.4 | -21.0±0.0 | -21.0±0.0 | 1.2±0.8 | 19.6±2.1 | 12.2±6.3 | 6.2±1.9 |
| **SA-DQN** | -21.0±0.0 | -20.8±0.4 | -20.8±0.4 | -21.0±0.0 | 19.4±3.0 | 27.6±0.5 | 19.4±1.5 | 19.6±2.2 |
| **WocaR-DQN** | -21.0±0.0 | -21.0±0.0 | -21.0±0.0 | -21.0±0.0 | 20.2±0.8 | 26.8±1.3 | 20.2±1.3 | 20.8±0.4 |
| **CAR-DQN** | -20.8±0.4 | -20.0±0.7 | -21.0±0.0 | -21.0±0.0 | 10.2±1.3 | 20.8±0.8 | 15.0±1.0 | 17.2±1.9 |
| **DP-DQN** | -20.6±0.5 | -9.2±4.5 | -21.0±0.0 | -21.0±0.0 | 21.8±3.2 | 28.0±1.2 | 10.6±2.5 | 7.4±1.3 |
| **DMBP** | -14.0±4.9 | -9.8±4.5 | -20.6±0.9 | -20.2±0.8 | 22.0±0.7 | 31.0±1.2 | 19.2±2.9 | 14.6±2.5 |
| | Doom | | | | Airsim | | | |
| **Model** | **SHIFT-O-1.0** | **SHIFT-O-0.25** | **SHIFT-I-1.0** | **SHIFT-I-0.25** | **SHIFT-O-1.0** | **SHIFT-O-0.25** | **SHIFT-I-1.0** | **SHIFT-I-0.25** |
| **DQN-No Attack** | 75.4±4.4 | 75.4±4.4 | 75.4±4.4 | 75.4±4.4 | 40.3±0.5 | 40.3±0.5 | 40.3±0.5 | 40.3±0.5 |
| **DQN** | -354.0±8.2 | 55.0±14.1 | -318.0±29.7 | -326.0±20.4 | 9.8±2.3 | 20.8±11.0 | 6.2±0.1 | 10.0±2.9 |
| **SA-DQN** | -375.0±13.2 | 62.2±4.6 | -337.0±43.1 | -30.8±206.4 | 5.4±0.4 | 6.7±1.6 | 3.7±0.5 | 8.6±2.5 |
| **WocaR-DQN** | -299.6±143.6 | -38.4±202.3 | -310.0±6.1 | -289.4±173.1 | 5.2±5.0 | 20.7±17.5 | 5.1±4.2 | 7.5±3.9 |
| **CAR-DQN** | -335.0±15.8 | 61.4±15.5 | -336.0±33.1 | -19.6±190.4 | 0.9±0.3 | 20.7±6.6 | 0.6±0.1 | 3.9±6.5 |
| **DP-DQN** | -116.4±173.2 | 64.0±10.6 | -303.0±2.7 | -252.0±172.9 | 18.8±2.3 | 17.8±8.8 | 4.8±3.1 | 10.5±3.6 |
| **DMBP** | -184.8±231.7 | 65.4±9.8 | -302.0±2.7 | -256.2±180.4 | 20.8±12.0 | 22.8±8.0 | 6.2±0.7 | 9.4±4.9 |

## 4   Experiments

We evaluate SHIFT using four Atari environments [7], Doom game [52] and Airsim [43] autonomous driving simulator. We consider state-of-the-art defenses including SA-DQN [60], WocaR-DQN [36], CAR-DQN [35], and two diffusion-based defenses: DMBP [57], which is a test-stage defense, where the victim uses a diffusion model conditioned on perturbed history to recover true states, and

Table 2: Comparison with different attack methods. We compare our SHIFT attack with PGD, MinBest, and PA-AD with {1/255,15/255} budgets and rotation (degree 1) and transform attacks, and report reward, deviation rate, reconstruction error, Wasserstein distance, SSIM, and LPIPS against DP-DQN.

| Env | Freeway | | | | | |
|---|---|---|---|---|---|---|
| Attack Method | Reward↓ | Dev (%)↑ | Recons.↓ | Wass. ($\times 10^{-3}$)↓ | SSIM↑ | LPIPS↓ |
| PGD-1/255 | $30.0 \pm 0.9$ | $3.5 \pm 0.2$ | $3.45 \pm 0.3$ | $0.81 \pm 0.1$ | $0.9972 \pm 0.0001$ | $0.0018 \pm 0.0005$ |
| PGD-15/255 | $29.0 \pm 1.0$ | $3.2 \pm 0.5$ | $4.36 \pm 0.3$ | $1.63 \pm 1.1$ | $0.7461 \pm 0.0086$ | $0.1669 \pm 0.022$ |
| MinBest-1/255 | $30.2 \pm 1.3$ | $3.7 \pm 0.3$ | $3.45 \pm 0.3$ | $0.88 \pm 0.2$ | $0.9965 \pm 0.0002$ | $0.0023 \pm 0.0006$ |
| MinBest-15/255 | $29.4 \pm 1.2$ | $7.3 \pm 0.2$ | $5.35 \pm 0.2$ | $1.75 \pm 0.6$ | $0.6555 \pm 0.0067$ | $0.2155 \pm 0.023$ |
| PA-AD-1/255 | $30.8 \pm 1.0$ | $6.5 \pm 0.1$ | $3.47 \pm 0.3$ | $0.82 \pm 0.2$ | $0.9957 \pm 0.0001$ | $0.0027 \pm 0.0007$ |
| PA-AD-15/255 | $29.0 \pm 1.1$ | $10.3 \pm 1.0$ | $6.06 \pm 0.2$ | $1.34 \pm 0.5$ | $0.5891 \pm 0.0064$ | $0.2368 \pm 0.0232$ |
| PA-AD-TC-15/255 | $28.5 \pm 1.2$ | $9.2 \pm 0.8$ | $6.01 \pm 0.2$ | $1.63 \pm 0.2$ | $0.5962 \pm 0.0054$ | $0.2373 \pm 0.0235$ |
| Rotation 1 Degree | $27.2 \pm 0.7$ | $26.9 \pm 0.5$ | $6.41 \pm 0.2$ | $0.80 \pm 0.2$ | $0.8237 \pm 0.0022$ | $0.0351 \pm 0.0041$ |
| Transform(1,0) | $26.8 \pm 0.4$ | $22.1 \pm 1.0$ | $9.32 \pm 0.1$ | $0.80 \pm 0.2$ | $0.6045 \pm 0.0074$ | $0.0741 \pm 0.0068$ |
| **SHIFT-O-1.0** | $\mathbf{21.8 \pm 3.2}$ | $\mathbf{22.8 \pm 2.6}$ | $\mathbf{1.02 \pm 0.5}$ | $\mathbf{0.89 \pm 0.3}$ | $\mathbf{0.9990 \pm 0.0014}$ | $\mathbf{0.0008 \pm 0.0014}$ |
| **SHIFT-I-1.0** | $\mathbf{10.6 \pm 2.5}$ | $\mathbf{51.4 \pm 3.9}$ | $\mathbf{1.05 \pm 0.5}$ | $\mathbf{0.84 \pm 0.2}$ | $\mathbf{0.9904 \pm 0.0043}$ | $\mathbf{0.0175 \pm 0.0124}$ |

DP-DQN [46], which uses a diffusion-based denoiser on top of a pre-trained pessimistic policy. We set the history length $k = 4$ for the two history-based defenses and our attacks. All other hyper-parameters of are given in Appendix C.7. We compare SHIFT with $l_\infty$-norm constrained PGD [60], MinBest [24], and PA-AD [47] attacks in the Atari Freeway environment, as well as two high-sensitivity direction-based attacks (rotation and transform) from Korkmaz [31]. The PGD attack aims to force the victim into taking non-optimal actions, while the MinBest attack minimizes the logit of the best action. PA-AD and its temporally coupled version [37] (PA-AD-TC) use RL to determine the best attack direction. Rotation/transform attack rotates/shifts the state by a given amount.

We evaluate our method using six metrics from four perspectives: 1) Attack Effectiveness: **Reward** (average episode return over 10 runs) and **Deviation Rate** (percentage of steps where the victim's action $\pi(\tilde{s}_t)$ differs from $\pi(s_t)$); 2) Perturbed States Realism: **Reconstruction Error** ($l_2$ distance $\|\tilde{s}_t - \mathbf{AE}(\tilde{s}_t)\|_2$ via realism autoencoder detector); 3) Historical Alignment: **Wasserstein-1 distance** between two consecutive perturbed states, which measures the cost of moving pixel mass and better represent image manipulations than the $l_p$ distance. Wong et al. [51] has used it to generate beyond $l_p$ norm constraint adversarial examples in supervised learning; 4) Trajectory Faithfulness: **SSIM** [50] and **LPIPS** [62] (perceptual similarity between $\tilde{s}_t$ and $s_t$; unavailable to the agent during testing, so SSIM and LPIPS cannot be computed). Additional results are provided in Appendix D, including results on a continuous action space environment using the PPO policy, detectability results on various attack methods, ablation studies on DDPM vs. EDM diffusion architectures, time complexity comparison of different attacks, diffusion model generated images detection results and a comparison with high-sensitivity direction attacks [31].

## 4.1 Main Results

**Attack Effectiveness.** Table 1 presents the performance of **SHIFT-O** and **SHIFT-I** under attack frequencies $\xi = 1.0$ and $0.25$ against a variety of defense mechanisms. Table 2 further reports metrics from four perspectives, comparing baseline attacks and our SHIFT variants against the DP-DQN defense. Our results show that both variants significantly degrade the return of the vanilla DQN agent and that under regularization-based defenses, even under low attack frequency. Although diffusion-based defenses outperform other baselines due to their ability to denoise using history-conditioned signals, our SHIFT attacks still induce substantial reward drops and high deviation rates. In contrast, baseline attacks are largely ineffective (shown in Table 2) because they fail to introduce semantically meaningful changes. **SHIFT-O** bypasses diffusion-based defenses by generating perturbed states that manipulate environment-specific semantics, while **SHIFT-I** constructs plausible imaginary trajectories that lead the agent to take suboptimal actions in the true environment.

**Attack Stealthiness.** In addition to achieving stronger attack performance, both SHIFT variants exhibit higher stealthiness across all evaluation metrics compared to baselines, highlighting the effectiveness of using conditional diffusion to generate stealthy yet successful attacks. Our attacks generate the most realistic perturbed states compared to baseline attacks, as evidenced by the lowest average reconstruction errors. They also achieve comparable historical-alignment and trajectory-faithfulness to $l_p$-norm-based attacks with small attack budgets and are better than those attacks with large budgets. **SHIFT-I** achieves stronger overall attack effectiveness than **SHIFT-O**, but has lower SSIM and higher LPIPS, indicating less trajectory faithfulness. However, it exhibits better temporal

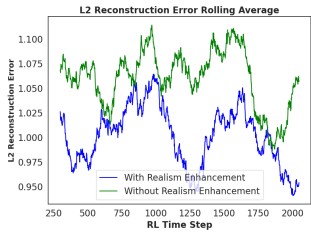
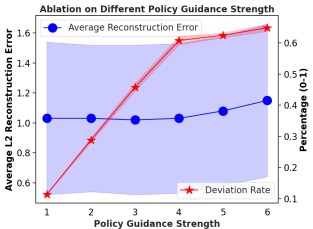
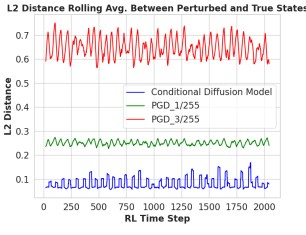

| (a) Realism Guidance | (b) Policy Guidance Strength | (c) $l_2$ Distance |

Figure 3: Ablation Study Results. a) shows the rolling average of $l_2$ reconstruction error (from the autoencoder-based realism detector) of our generated perturbed states with and without the realism enhancement. b) shows the $l_2$ reconstruction error, deviation rate under different policy guidance strengths with **SHIFT-O** attack. c) shows distance between perturbed and true states ($84 \times 84$ grayscale images). (a) and (b) use the vanilla DQN policy.

consistency, reflected by a lower average Wasserstein distance. Lastly, our results in Table 2 support the presence of a fundamental *trilemma*: no attack dominates across all key metrics simultaneously.

## 4.2 Ablation Studies

**Ablation on Realism Guidance.** Figure 3a illustrates the $l_2$ reconstruction error, which is defined as $\|s_t - \mathbf{AE}(s_t)\|_2$, for generated perturbed states both with and without the realism enhancement component. The figure demonstrates that, by incorporating realism enhancement, the $l_2$ reconstruction error is significantly reduced. This reduction indicates that realism enhancement effectively contributes to the generation of perturbed states that are more stealthy and less likely to be detected.

**Ablation on Policy Guidance Strength** Figure 3b illustrates the performance of our attack across various levels of policy guidance strength $\Gamma_2$. The figure indicates that as the strength increases, the effectiveness of our attack improves, leading to higher manipulation and deviation rates. However, this increased strength also results in a higher $l_2$ reconstruction error, which negatively impacts the realism of the generated perturbed states. Consequently, there exists a trade-off between attack effectiveness and stealthiness when selecting different policy guidance strengths.

## 5 Conclusion and Future Work

We introduce a novel policy-agnostic diffusion-based state perturbation attack for reinforcement learning (RL) systems that extends beyond the traditional $l_p$-norm constraints. By leveraging conditional diffusion models, policy guidance, and realism enhancement techniques, we generate highly effective, semantically distinct, and stealthy attacks that cause a significant reduction in cumulative rewards across multiple environments. Our results underscore the urgent need for more sophisticated defense mechanisms to effectively mitigate semantic uncertainties. We also propose a fundamental *trilemma* in adversarial perturbation attacks in the image domain: any existing attack method can only optimize two out of three critical properties—**Semantic Change**, **Historical Alignment**, and **Trajectory Faithfulness**. Finding an attack that better satisfies all three properties than current methods remains an open problem.

## Acknowledgments

This work has been funded in part by NSF grant CNS-2146548. We thank the anonymous reviewers for their insightful feedback.

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

# Appendix

## Impact Statement

Our work introduces SHIFT, a novel approach to perturbation attacks in reinforcement learning (RL) by altering the semantics of the true states while remaining stealthy from both static and dynamic perspectives. SHIFT demonstrates outstanding performance, successfully compromising all state-of-the-art defense methods. This highlights the urgent need for more sophisticated defense mechanisms that are resilient to semantic uncertainties.

This research raises important safety concerns, as adversaries could exploit these semantic-changing attacks to cause significant harm in real-world RL applications, such as autonomous driving. The ability to alter the perception of critical systems like self-driving cars could lead to catastrophic consequences. As such, our findings underscore the necessity of further research into robust defenses capable of withstanding such advanced and subtle attack strategies.

# A Related Work

## A.1 State Perturbation Attacks and Defenses

State perturbation attacks on RL policies were first introduced in [24], where the *MinBest* attack was proposed to minimize the probability of selecting the best action by iteratively adding $l_p$-norm constrained noise calculated through $-\nabla_{\tilde{s}_t}\pi(\pi(\cdot|s_t), \tilde{s}_t)$. Building on this, Zhang et al. [60] showed that when the agent's policy is fixed, finding the optimal adversarial policy can be framed as an MDP, and the attacker can find the optimal attack policy by applying RL techniques. This was further improved in [47], where a more efficient algorithm for finding optimal attacks, called *PA-AD*, was introduced. Instead of searching perturbed states in the original state space, *PA-AD* trained a director through RL to find the optimal attack direction, and the trained director directs the designed actor to generate perturbed states, which decreases the searching space of RL. More recently, illusory attack [16] is proposed by requiring a perturbed trajectory to follow the same distribution as the normal trajectory, making it difficult to detect. However, this approach does not scale to high-dimensional image input. Korkmaz [31] recognized the limitation of $l_p$ norm constrained attacks and proposed a policy-independent attack by following high sensitive directions, leading to attacks such as changing brightness and contrast, image blurring, image rotation and image transform. These types of attacks are imperceptible when the amount of manipulation applied is small and can compromise SA-MDP defense. However, they can barely alter the domain-specific semantics of image input according to our Definition 3.3. For example, in the Pong game, the pong ball's relative distance from the two pads will remain the same after changing brightness and contrast, or transforming the image. As a result, these attacks cannot bypass diffusion-based defenses as shown in Tables 8 and 9.

On the defense side, Zhang et al. [60] demonstrated that a universally optimal policy under state perturbations might not always exist. They proposed a set of regularization-based algorithms (SA-DQN, SA-PPO, SA-DDPG) to train robust RL agents. This was enhanced in [36], where a worst-case Q-network and state importance weights were incorporated into the regularization. A more recent work called CAR-DQN [35] shows that using an $l_\infty$ norm can further improve the policy's robustness, and they theoretically capture the optimal robust policy (ORP) under $\epsilon$ constrained state perturbation attacks, although this method incurs high computational costs. Another line of work by [55] proposed an autoencoder-based detection and denoising framework to identify and correct perturbed states. Korkmaz and Brown-Cohen [32] proposed SO-INRD, which uses the local curvature of the cross-entropy loss between the action distribution $\pi(a|s)$ given by a policy $\pi$ and a target action distribution to detect adversarial directions. He et al. [20] showed that when the initial state distribution is known, it is possible to find a policy that optimizes the expected return under state perturbations. Diffusion-based defenses have also been utilized to generate more robust agent policies. DMBP [57] utilized a conditional diffusion model to recover actual states from perturbed states and Sun and Zheng [46] used the diffusion model as a purification tool to generate a belief set about the actual state and perform a pessimistic training to generate a robust policy. Nie et al. [40] found that the performance loss of a policy under an $l_\infty$ norm-bounded state perturbations is bounded by the KL divergence between the action distribution under the true state and that under the perturbed state, and utilized a new network architecture called SortNet [59] to train a robust RL policy. More recently, a game-theoretical defense method (GRAD) [37] was proposed to address temporally coupled attacks by modeling the temporally coupled robust RL problem as a partially observable zero-sum game and deriving an approximate equilibrium of the game. Another important recent defense is PROTECTED [39], which iteratively searches for a set of non-dominated policies during training and adapts these policies during testing to address different attacks. However, both GRAD [37] and PROTECTED [39] focus on MuJoCo environments and are already computationally intensive (both take more than 20 hours) to train on the relatively simple environments. Without further adaptation, it will be computationally prohibitive to apply these two methods to environments with image input as we consider in this paper.

## A.2 Attacks and Defenses Beyond State Perturbations

As demonstrated by [25], altering the reward signal can significantly disrupt the training process of Q-learning, causing the agent to adopt a policy that aligns with the attacker's objectives. Additionally, [63] introduced an adaptive reward poisoning technique that can induce a harmful policy in a number of steps that scales polynomially with the size of the state space $|S|$ in the tabular setting. In

a similar vein, Zhang et al. [63] developed an adaptive reward poisoning method capable of achieving a malicious policy in polynomial steps based on the size of the state space $|S|$.

Moving beyond reward manipulation, Lee et al. [33] proposed two techniques for perturbing the action space. Among them, the *Look-Ahead Action Space* (LAS) method was found to deliver better performance in reducing cumulative rewards in deep reinforcement learning by distributing attacks across both the action and temporal dimensions. Another line of research focuses on adversarial policies within multi-agent environments. For example, [17] showed that in a zero-sum game, a player using an adversarial policy can easily beat an opponent using a well-trained policy.

Attacks targeting an RL agent's policy network have also been explored. Inference attacks, as described by [10], aim to steal the policy network parameters. On the other hand, poisoning attacks, as discussed in [23], focus on directly manipulating the model parameters. Specifically, Huai et al. [23] proposed an optimization-based method to identify an optimal strategy for poisoning the policy network.

### A.3 Diffusion Models and RL

Diffusion models have recently been utilized to solve RL problems by exploiting their state-of-the-art sample generation ability. In particular, diffusion models have been utilized to generate high quality offline data in solving offline RL problems. Offline RL training is known as a data-sensitive process, where the quality of the data has a huge influence on the training result. To deal with this problem, many studies [19, 2, 27] have shown that diffusion models can learn from a demo dataset and then generate high reward trajectories for learning or planning purposes. In addition, conditional diffusion models have been directly used to model RL policies. A conditional diffusion model can generate actions through a denoising process with states and other useful information as conditions. Several studies [28, 48] have shown state-of-the-art performance in various offline RL environments when using a diffusion model as a policy, which leads to a promising research direction.

Furthermore, Black et al. [8] shows that the denoising process can be viewed as a Markov Decision Process (MDP). Thus, Black et al. [8] trains a diffusion model with the help of RL by maximizing a user-specific reward function, which connects the generative models and optimization methods.

### A.4 Diffusion Models in Adversarial Examples

Diffusion models have recently gained significant attention in generating adversarial examples due to their superb performance. They can generate high-quality adversarial examples that deceive target classifiers while remaining imperceptible to human observers.

Since the images generated by diffusion models inherently lack adversarial effects, a widely used approach is to use diffusion models along with existing methods of generating adversarial examples. The idea is to combine the generated samples from the diffusion model with perturbed samples from other attack methods such as PGD attacks during the attack process to generate high quality and imperceptible adversarial examples [56, 11].

Another promising direction is to use a (surrogate) classifier to guide the diffusion model generating samples that meet attacker specified goals by using gradient information from the classifier during the testing stage [38, 13, 18]. Also, Chen et al. [9] used the classifier guidance during the training stage of the diffusion model along with self and cross attention mechanisms.

Further, Beerens et al. [6] showed that poisoning the training set can produce a deceptive diffusion model that will generate adversarial samples without any guidance.

However, these works only care about static stealthiness in a supervised learning setting, while SHIFT also takes dynamic stealthiness into consideration.

### A.5 Relation with Constrained Diffusion Model

While diffusion models have been utilized to generate adversarial examples in the supervised learning setting (see Appendix A.4 for a review), their application in adversarial state perturbations in RL has not been considered before. We remark that our problem can be viewed as sampling from a diffusion model with constraints on realism and history alignment. However, existing approaches for

constrained diffusion [12] cannot be applied directly to our setting, as they require constraints such as physical rules to be explicitly given and easily evaluable. In our setting, it is difficult to identify the projection onto the valid states $S^*$, making these approaches less suitable.

### A.6 GAN-based Semantic Adversarial Attacks.

Generative adversarial networks (GANs) have been widely used to generate adversarial examples that go beyond traditional $_p$-norm constraints in image classification. Notable works include Adv-GAN [53], which learns to generate perturbations that mislead classifiers while maintaining perceptual similarity, and Natural Adversarial Examples [64], which leverage latent space manipulations within VAE-GAN frameworks to produce realistic semantic shifts. Other approaches explore latent adversarial attacks [65] or direct image synthesis [4] to create imperceptible or highly transferable adversarial examples. While these methods succeed in the supervised setting, they do not transfer directly to reinforcement learning due to the absence of sequential structure, policy conditioning, or temporal consistency in GANs. Moreover, aligning GANs or other generative methods with reinforcement learning (RL)-specific objectives—such as minimizing the expected cumulative reward under a given policy—would require substantial modifications to both architecture and loss functions. As such, adapting these methods to generate semantics-aware perturbations in RL constitutes a distinct line of research and is not directly comparable to our approach.

## B  Proof of Theorem 3.6

The following proof is adapted from the proof in Appendix H of Dhariwal and Nichol [14]. We show that in the RL state perturbation attacks setting, we could combine classifier-free and gradient guidance. Let $\pi$ denote the victim's policy, $Q^\pi$ the state-action value function, and $\tau_{t-1} = \{s_{t-1}, a_{t-1}, ..., s_{t-k}, a_{t-k}\}$ the sequence of the last $k$ observations and actions up to time $t$. We first define a conditional Markovian process $\hat{q}$ similar to $q$ as follows.

$$
\begin{aligned}
&\hat{q}(\tilde{s}_t^0 | \tau_{t-1}) := q(\tilde{s}_t^0 | \tau_{t-1}) \\
&\hat{q}(Q^\pi | \tilde{s}_t^0, \tau_{t-1}) \text{ is known for every } (\tilde{s}_t^0, \tau_{t-1}) \\
&\hat{q}(\tilde{s}_t^{i+1} | \tilde{s}_t^i, Q^\pi, \tau_{t-1}) := q(\tilde{s}_t^{i+1} | \tilde{s}_t^i), \quad \forall i \\
&\hat{q}\left(\tilde{s}_t^{1:T} \mid \tilde{s}_t^0, Q^\pi, \tau_{t-1}\right) := \prod_{i=1}^T \hat{q}\left(\tilde{s}_t^i \mid \tilde{s}_t^{i-1}, Q^\pi, \tau_{t-1}\right),
\end{aligned}
\tag{4}
$$

where $q(\tilde{s}_t^0 | \tau_{t-1}) = P(\tilde{s}_t^0 | s_{t-1}, a_{t-1})$ is the conditional distribution of the original state $\tilde{s}_t^0$ given the history $\tau_{t-1}$. Next we show that the joint distribution $\hat{q}(\tilde{s}_t^{0:T}, Q^\pi | \tau_{t-1})$ given $\tau_{t-1}$ is well defined.

$$
\begin{aligned}
\hat{q}(\tilde{s}_t^{0:T}, Q^\pi | \tau_{t-1}) &= \hat{q}\left(\tilde{s}_t^{1:T} \mid \tilde{s}_t^0, Q^\pi, \tau_{t-1}\right) \hat{q}(\tilde{s}_t^0, Q^\pi | \tau_{t-1}) \\
&= \prod_{i=1}^T \hat{q}\left(\tilde{s}_t^i \mid \tilde{s}_t^{i-1}, Q^\pi, \tau_{t-1}\right) \hat{q}(Q^\pi | \tilde{s}_t^0, \tau_{t-1}) \hat{q}(\tilde{s}_t^0 | \tau_{t-1}) \\
&= \prod_{i=1}^T \hat{q}\left(\tilde{s}_t^i \mid \tilde{s}_t^{i-1}, Q^\pi, \tau_{t-1}\right) \hat{q}(Q^\pi | \tilde{s}_t^0, \tau_{t-1}) \hat{q}(\tilde{s}_t^0 | \tau_{t-1}) \\
&= \prod_{i=1}^T \hat{q}\left(\tilde{s}_t^i \mid \tilde{s}_t^{i-1}, Q^\pi, \tau_{t-1}\right) \hat{q}(Q^\pi | \tilde{s}_t^0, \tau_{t-1}) \hat{q}(\tilde{s}_t^0 | \tau_{t-1}).
\end{aligned}
$$

Following essentially the same reasoning as in Appendix H of  Dhariwal and Nichol [14] with the trivial extension of including the condition $\tau_{t-1}$, we have

$$
\begin{aligned}
\hat{q}(\tilde{s}_t^i | \tilde{s}_t^{i-1}, \tau_{t-1}) &= \hat{q}(\tilde{s}_t^i | \tilde{s}_t^{i-1}) \\
\hat{q}(\tilde{s}_t^{i-1} | \tilde{s}_t^i, \tau_{t-1}) &= q(\tilde{s}_t^{i-1} | \tilde{s}_t^i, \tau_{t-1})
\end{aligned}
$$

Next, we show $\hat{q}(Q^\pi|\tilde{s}_t^i, \tilde{s}_t^{i-1}, \tau_{t-1})$ does not depend on $\tilde{s}_t^i$.

$$
\begin{aligned}
\hat{q}(Q^\pi|\tilde{s}_t^i, \tilde{s}_t^{i-1}, \tau_{t-1}) &= \frac{\hat{q}(\tilde{s}_t^{i-1}, \tilde{s}_t^i, Q^\pi, \tau_{t-1})}{\hat{q}(\tilde{s}_t^i, \tilde{s}_t^{i-1}, \tau_{t-1})} \\
&= \hat{q}(\tilde{s}_t^i|\tilde{s}_t^{i-1}, Q^\pi, \tau_{t-1})\frac{\hat{q}(\tilde{s}_t^{i-1}, Q^\pi, \tau_{t-1})}{\hat{q}(\tilde{s}_t^{i-1}, \tilde{s}_t^i, \tau_{t-1})} \\
&= \hat{q}(\tilde{s}_t^i|\tilde{s}_t^{i-1})\frac{\hat{q}(Q^\pi|\tilde{s}_t^{i-1}, \tau_{t-1})}{\hat{q}(\tilde{s}_t^i|\tilde{s}_t^{i-1}, \tau_{t-1})} \\
&= \hat{q}(\tilde{s}_t^i|\tilde{s}_t^{i-1})\frac{\hat{q}(Q^\pi|\tilde{s}_t^{i-1}, \tau_{t-1})}{\hat{q}(\tilde{s}_t^i|\tilde{s}_t^{i-1})} \\
&= \hat{q}(Q^\pi|\tilde{s}_t^{i-1}, \tau_{t-1}).
\end{aligned}
\tag{5}
$$

We can now derive the reverse process that combines both classifier-free and gradient-guided methods.

$$
\begin{aligned}
\hat{q}(\tilde{s}_t^{i-1}|\tilde{s}_t^i, Q^\pi, \tau_{t-1}) &= \frac{\hat{q}(\tilde{s}_t^{i-1}, \tilde{s}_t^i, Q^\pi, \tau_{t-1})}{\hat{q}(\tilde{s}_t^i, Q^\pi, \tau_{t-1})} \\
&= \frac{\hat{q}(\tilde{s}_t^{i-1}, \tilde{s}_t^i, \tau_{t-1})\hat{q}(Q^\pi|\tilde{s}_t^{i-1}, \tilde{s}_t^i, \tau_{t-1})}{\hat{q}(\tilde{s}_t^i, Q^\pi, \tau_{t-1})} \\
&= \frac{\hat{q}(\tilde{s}_t^{i-1}|\tilde{s}_t^i, \tau_{t-1})\hat{q}(\tilde{s}_t^i, \tau_{t-1})\hat{q}(Q^\pi|\tilde{s}_t^{i-1}, \tilde{s}_t^i, \tau_{t-1})}{\hat{q}(\tilde{s}_t^i, Q^\pi, \tau_{t-1})} \\
&= \frac{\hat{q}(\tilde{s}_t^{i-1}|\tilde{s}_t^i, \tau_{t-1})\hat{q}(Q^\pi|\tilde{s}_t^{i-1}, \tilde{s}_t^i, \tau_{t-1})}{\hat{q}(Q^\pi|\tilde{s}_t^i, \tau_{t-1})} \\
&\overset{(a)}{=} q(\tilde{s}_t^{i-1}|\tilde{s}_t^i, \tau_{t-1})\frac{\hat{q}(Q^\pi|\tilde{s}_t^{i-1}, \tau_{t-1})}{\hat{q}(Q^\pi|\tilde{s}_t^i, \tau_{t-1})},
\end{aligned}
$$

where (a) follows from Equation (5). Note that $q(\tilde{s}_t^{i-1}|\tilde{s}_t^i, \tau_{t-1})$ can be learned through a history conditioned diffusion model $p_\theta$ and we will use $Q^\pi(s_t, \pi(\tilde{s}_t^i))$ to approximate $\hat{q}(Q^\pi|\tilde{s}_t^i, \tau_{t-1})$, $\forall i$. We notice that the $Q^\pi(s_t, \pi(\tilde{s}_t^i))$ term guides our reverse process to generate samples that lead to a lower state value. Thus, the attacker should use the victim's policy $\pi$ here to gain better guidance through $Q^\pi(\tilde{s}_t^i, \pi(\tilde{s}_t^i))$. Plugging them back into the above equation, we have

$$
\begin{aligned}
\hat{q}(\tilde{s}_t^{i-1}|\tilde{s}_t^i, Q^\pi, \tau_{t-1}) &\approx p_\theta(\tilde{s}_t^{i-1}|\tilde{s}_t^i, \tau_{t-1})\frac{Q^\pi(s_t, \pi(\tilde{s}_t^{i-1}))}{Q^\pi(s_t, \pi(\tilde{s}_t^i))} \\
&\approx p_\theta(\tilde{s}_t^{i-1}|\tilde{s}_t^i, \tau_{t-1})e^{\log Q^\pi(s_t, \pi(\tilde{s}_t^{i-1})) - \log Q^\pi(s_t, \pi(\tilde{s}_t^i))}.
\end{aligned}
\tag{6}
$$

Using the Taylor expansion, we get

$$
\log Q^\pi(s_t, \pi(\tilde{s}_t^{i-1})) - \log Q^\pi(s_t, \pi(\tilde{s}_t^i)) \approx (\tilde{s}_t^{i-1} - \tilde{s}_t^i)\nabla_{\tilde{s}_t^i}\log Q^\pi(s_t, \pi(\tilde{s}_t^i)).
$$

We also have

$$
p_\theta(\tilde{s}_t^{i-1}|\tilde{s}_t^i, \tau_{t-1}) \propto \mathcal{N}(\tilde{s}_t^{i-1}; \mu_i, i), \sigma_i^2\mathbf{I}) \propto \exp\left(-\frac{\left(\tilde{s}_t^{i-1} - \mu_i\right)^2}{2\sigma_i^2}\right).
$$

where $\mu_i$ comes from $\epsilon_i = \Gamma\epsilon_\theta(s_t^i, i, \tau_{t-1}) + (1 - \Gamma)\epsilon_\theta(s_t^i, i)$, as given by (2), and $\sigma_i^2$ is determined by the noise scheduler $\beta_i$.

Substituting them back into (6), we have

$$p_\theta(\tilde{s}_t^{i-1}|\tilde{s}_t^i, \tau_{t-1})e^{\log Q^\pi(s_t, \pi(\tilde{s}_t^{i-1})) - \log Q^\pi(s_t, \pi(\tilde{s}_t^i))}$$

$$\propto \exp\left(-\frac{\left(\tilde{s}_t^{i-1} - \mu_i\right)^2}{2\sigma_i^2} + (\tilde{s}_t^{i-1} - \tilde{s}_t^i)\nabla_{\tilde{s}_t^i}\log Q^\pi(s_t, \pi(\tilde{s}_t^i))\right)$$

$$= \exp\left(-\frac{\left(\tilde{s}_t^{i-1} - \mu_i\right)^2 - 2\sigma_i^2\left(\tilde{s}_t^{i-1} - \tilde{s}_t^i\right)\nabla_{\tilde{s}_t^i}\log Q^\pi(s_t, \pi(\tilde{s}_t^i))}{2\sigma_i^2}\right)$$

$$= \exp\left(-\frac{\left(\tilde{s}_t^{i-1} - \mu_i\right)^2 - 2\sigma_i^2\left(\tilde{s}_t^{i-1} - \mu_i\right)\nabla_{\tilde{s}_t^i}\log Q^\pi(s_t, \pi(\tilde{s}_t^i)) + \left(\sigma_i^2\nabla_{\tilde{s}_t^i}\log Q^\pi(s_t, \pi(\tilde{s}_t^i))\right)^2}{2\sigma_i^2}\right)$$

$$\times \exp\left(\frac{2\sigma_i^2\left(\mu_i - \tilde{s}_t^i\right)\nabla_{\tilde{s}_t^i}\log Q^\pi(s_t, \pi(\tilde{s}_t^i)) + \left(\sigma_i^2\nabla_{\tilde{s}_t^i}\log Q^\pi(s_t, \pi(\tilde{s}_t^i))\right)^2}{2\sigma_i^2}\right)$$

$$= \exp\left(-\frac{\left(\left(\tilde{s}_t^{i-1} - \mu_i\right) - \sigma_i^2\nabla_{\tilde{s}_t^i}\log Q^\pi(s_t, \pi(\tilde{s}_t^i))\right)^2}{2\sigma_i^2}\right.$$

$$\left. +\frac{2\sigma_i^2\left(\mu_i - \tilde{s}_t^i\right)\nabla_{\tilde{s}_t^i}\log Q^\pi(s_t, \pi(\tilde{s}_t^i)) + \left(\sigma_i^2\nabla_{\tilde{s}_t^i}\log Q^\pi(s_t, \pi(\tilde{s}_t^i))\right)^2}{2\sigma_i^2}\right)$$

$$\propto \exp\left(-\frac{\left(\tilde{s}_t^{i-1} - \left(\mu_i + \sigma_i^2\nabla_{\tilde{s}_t^i}\log Q^\pi(s_t, \pi(\tilde{s}_t^i))\right)\right)^2}{2\sigma_i^2}\right). \tag{7}$$

Equation (7) implies that the reverse process when sampling from a history-conditioned DDPM model guided by the victim's state value function can be represented as

$$p(\tilde{s}_t^{i-1}|\tilde{s}_t^i, Q^\pi, \tau_{t-1}) = \mathcal{N}\left(\tilde{s}_t^{i-1}; \mu_i + \sigma_i^2\nabla_{\tilde{s}_t^i}\log Q^\pi(s_t, \pi(\tilde{s}_t^i)), \sigma_i^2\mathbf{I}\right).$$

Noticed that we want to guide the generated state $\tilde{s}_t^0$ lead to a lower state-action value, thus we change sign of the gradient guidance to minus.

$$p(\tilde{s}_t^{i-1}|\tilde{s}_t^i, Q^\pi, \tau_{t-1}) = \mathcal{N}\left(\tilde{s}_t^{i-1}; \mu_i - \sigma_i^2\nabla_{\tilde{s}_t^i}\log Q^\pi(s_t, \pi(\tilde{s}_t^i)), \sigma_i^2\mathbf{I}\right).$$

# C  Implementation Details and Algorithms

## C.1  Two Stage Attacks Pipelines

Figure 4 gives an overview of our two-stage diffusion-based attack including all the major components involved.

## C.2  Visualization of SHIFT-O and SHIFT-I in the Freeway Environment and Doom

Figure 6 gives an illustration of SHIFT-O and SHIFT-I in the Atari Freeway environment. Figure 5 provides additional comparison between SHIFT-O/I and PGD and rotation attack from Korkmaz [31] in Atari Freeway and Doom.

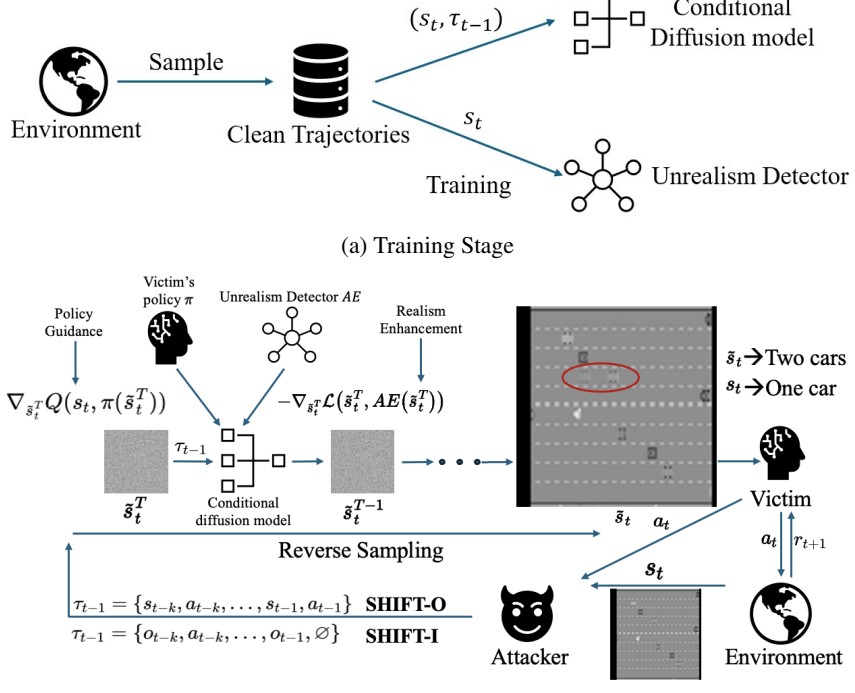

(a) Training Stage

(b) Testing Stage

Figure 4: Pipelines of SHIFT's two stages. a) shows the training stage where the attacker uses clean data to train a history-conditioned diffusion model and an autoencoder-based anomaly detector. b) shows the testing stage where the attacker perturbs the true state through the reverse sampling process of the pre-trained conditional diffusion model guided by the gradient of the victim's policy and that of the autoencoder's reconstruction loss.

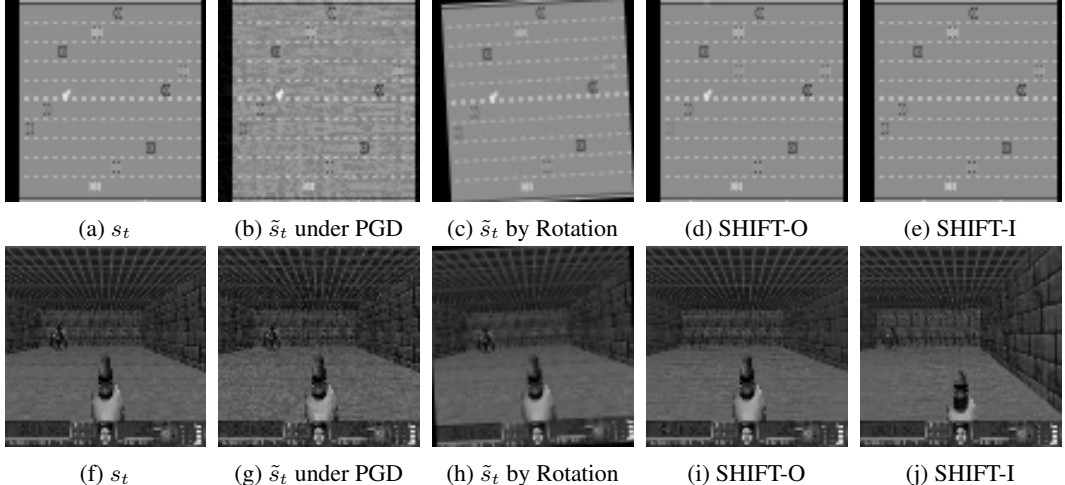

| (a) $s_t$ | (b) $\tilde{s}_t$ under PGD | (c) $\tilde{s}_t$ by Rotation | (d) SHIFT-O | (e) SHIFT-I |

| (f) $s_t$ | (g) $\tilde{s}_t$ under PGD | (h) $\tilde{s}_t$ by Rotation | (i) SHIFT-O | (j) SHIFT-I |

Figure 5: Extra examples of true and perturbed states of Atari Freeway and Doom. a) is the true state. b) and c) are the perturbed states under the PGD attack with a $l_\infty$ budget of $\frac{15}{255}$ and through rotation [31] by 3 degrees counterclockwise, respectively. d) and e) are the perturbed states generated by our SHIFT-O and SHIFT-I attacks, respectively. Neither PGD nor Rotation attacks can alter the decision-related semantics.

## C.3 Score-Based Diffusion Model

As shown in Song et al. [45], a diffusion process $\{x(i)\}_{i \in [0,T]}$ can be represented as the solution to a standard stochastic differential equation (SDE):

$$dx = f(x, i)di + g(i)dw,$$

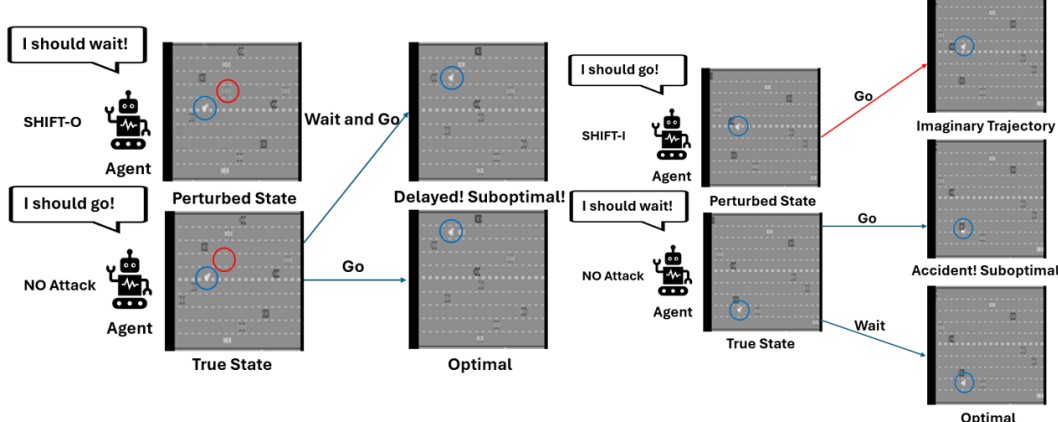

Figure 6: Visualization of **SHIFT-O** and **SHIFT-I** in Freeway. The goal of the Freeway game is to control the agent (blue circle) to cross the road quickly and safely. **SHIFT-O** injects an extra car (red circle) into the agent's observations to slow down the agent. **SHIFT-I** injects a segment of imaginary trajectory into the agent's observations that misleads the agent to move forward, despite a car being present in front of the agent in the real state, resulting in a collision.

where $\boldsymbol{f}$ represents the drift coefficient, which models the deterministic part of the SDE and determines the rate at which the process changes over time on average. $g(i)$ is called the diffusion coefficient, which represents the random part of the SDE and determines the magnitude of the noise. Finally, $\boldsymbol{w}$ represents a Brownian motion so that $g(i)d\boldsymbol{w}$ is the noising process.

We can let the diffusion process have $\boldsymbol{x}_0 \sim p_0$ and $\boldsymbol{x}_T \sim p_T$, where $p_0 = p_{\text{data}}$ is the data distribution and $p_T$ is a Gaussian noise distribution independent of $p_0$. Then we could run the reverse-time SDE to recover a sample from $p_0$ by the following process:

$$d\boldsymbol{x} = \left[\boldsymbol{f}(\boldsymbol{x}, i) - g(i)^2 \nabla_{\boldsymbol{x}} \log p_i(\boldsymbol{x})\right] di + g(i)d\overline{\boldsymbol{w}},$$

where $\nabla_{\boldsymbol{x}} \log p_i(\boldsymbol{x})$ is the score function and $\overline{\boldsymbol{w}}$ is a Brownian motion that flows back from time $T$ to $0$. The training objective for the score matching fucntion $\boldsymbol{s}_\theta$ for the SDE is then given by:

$$\arg\min_\theta \mathbb{E}_i \left[\lambda(i) \mathbb{E}_{\boldsymbol{x}(0)} \mathbb{E}_{\boldsymbol{x}(i)|\boldsymbol{x}(0)} \left[\left\|\boldsymbol{s}_\theta(\boldsymbol{x}(i), i) - \nabla_{\boldsymbol{x}(i)} \log p_{0i}(\boldsymbol{x}(i) \mid \boldsymbol{x}(0))\right\|_2^2\right]\right],$$

where $\lambda(i)$ is a positive weighting function and $i$ is uniformly sampled from $[0, T]$. The objective can be further simplified since $p_{0i}$ is a known Gaussian distribution.

### C.4 EDM Model and ODE formulation

Inspired by Song et al. [45], EDM [29] proposes an ODE formulation of the diffusion model by having a scheduler $\sigma(t)$ to schedule the noise added at each time step $t$. The score function correspondingly changes to $\nabla_{\boldsymbol{x}} \log p(\boldsymbol{x}; \sigma)$, which does not depend on the normalization constant of the underlying density function $p(\boldsymbol{x}, \sigma)$ and is much easier to evaluate. To be specific, if $D(\boldsymbol{x}; \sigma)$ is a denoiser function that minimizes:

$$\mathbb{E}_{\boldsymbol{x} \sim p_{\text{data}}} \mathbb{E}_{\boldsymbol{n} \sim \mathcal{N}(\boldsymbol{0}, \sigma^2 \mathbf{I})} \|D(\boldsymbol{x} + \boldsymbol{n}; \sigma) - \boldsymbol{x}\|_2^2 \tag{8}$$

then

$$\nabla_{\boldsymbol{x}} \log p(\boldsymbol{x}; \sigma) = (D(\boldsymbol{x}; \sigma) - \boldsymbol{x})/\sigma^2$$

We usually train a neural network $\theta$ to learn the denoising function $D(\boldsymbol{x}, \sigma)$ by using the simplified training objective in Equation (8). Utilizing this finding, EDM only requires a small number of reverse sampling steps to generate a high quality sample. However, EDM needs more preconditioning parameters such as scaling $\boldsymbol{x}$ to an approximate dynamic range, as further discussed below.

## C.5 EDM as a conditional diffusion model

In this paper, we follow the approach in Alonso et al. [3] to train an EDM-based diffusion model conditioned on a history $\tau_{t-1}$ to predict the next state $s_t$. Taking the network preconditioning parameters used in EDM into account, we have the denoising function changed to:

$$\mathbf{D}_\theta \left( s_t^i, c_{\text{noise}}^i, \tau_{t-1} \right) = c_{\text{skip}}^i \, s_t^i + c_{\text{out}}^i \, \mathbf{F}_\theta \left( c_{\text{in}}^i \, s_t^i, c_{\text{noise}}^i, \tau_{t-1} \right),$$

where $\mathbf{F}_\theta$ is the neural network to be trained, the preconditioners $c_{\text{in}}$ and $c_{\text{out}}$ scale the network's input and output magnitude to keep them at unit variance for any noise level $\sigma(i)$, $c_{\text{noise}}^i$ is an empirical transformation of the noise level, and $c_{\text{skip}}^i$ is determined by the noise level $\sigma(i)$ and the standard deviation of the data distribution $\sigma_{\text{data}}$. The detailed expressions are given below:

$$c_{\text{in}}^i = \frac{1}{\sqrt{\sigma(i)^2 + \sigma_{\text{data}}^2}} \tag{9}$$

$$c_{\text{out}}^i = \frac{\sigma(i)\sigma_{\text{data}}}{\sqrt{\sigma(i)^2 + \sigma_{\text{data}}^2}} \tag{10}$$

$$c_{\text{noise}}^i = \frac{1}{4}\log(\sigma(i)) \tag{11}$$

$$c_{\text{skip}}^i = \frac{\sigma_{\text{data}}^2}{\sigma_{\text{data}}^2 + \sigma^2(i)} \tag{12}$$

where $\sigma_{\text{data}} = 0.5$. The noise parameter $\sigma(i)$ is sampled to maximize the effectiveness during training by setting $\log(\sigma(i)) = \mathcal{N}(P_{\text{mean}}, P_{\text{std}}^2)$, where $P_{\text{mean}} = -0.4, P_{\text{std}} = 1.2$. Refer to Karras et al. [29] for a detailed explanation.

The training objective of $\mathbf{F}_\theta$ changes correspondingly to

$$\mathcal{L}(\theta) = \mathbb{E}[\|\mathbf{F}_\theta \left( c_{\text{in}}^i \, s_t^i, c_{\text{noise}}^i, \tau_{t-1} \right) - \frac{1}{c_{\text{out}}^i} \left( s_t - c_{\text{skip}}^i \, s_t^i \right) \|^2] \tag{13}$$

In our implementation, we change the residual block layers from [2,2,2,2] to [2,2] and the denosing steps to $5$, and set the drop condition rate to $0.1$. We keep other hyperparameters the same as Alonso et al. [3].

## C.6 Training and testing stage algorithms for SHIFT

---
**Algorithm 1:** History-Aligned Conditional Diffusion Model Training

---
**Input:** Training data $O = \{(s_t, \tau_{t-1})\}_{i=1}^N$, condition dropping rate $\alpha_{\text{drop}}$, $P_{\text{mean}}$, $P_{\text{std}}^2$, $\sigma$, learning rate $\eta$, history length $k$.
**Output:** Trained EDM model parameters $\theta$

**Initialize:** EDM model parameters $\theta$
**for** *number of training iterations* **do**
    Sample a data point $(s_t, \tau_{t-1}) \sim O$;
    // $\tau_{t-1} = \{s_{t-k}, a_{t-k}, \ldots, s_{t-1}, a_{t-1}\}$ is the **true** history
    Sample $\log(\sigma) \sim \mathcal{N}(P_{\text{mean}}, P_{\text{std}}^2)$;
    Calculate preconditioners $c_{\text{in}}, c_{\text{out}}$ based on $\sigma$ according to (9) and (10);
    Generate noisy data $s_t^i \sim \mathcal{N}(s_t, \sigma^2 \mathbf{I})$;
    **if** *random* $> \alpha_{drop}$ **then**
        | Compute generated state $\tilde{s}_t = \mathbf{D}_\theta(s_t^i, c_{\text{noise}}, \tau_{t-1})$
    **else**
        | Compute generated state $\tilde{s}_t = \mathbf{D}_\theta(s_t^i, c_{\text{noise}})$
    Compute loss $\mathcal{L}(\theta)$ based on Equation (13);
    Update $\theta$ using gradient descent: $\theta \leftarrow \theta - \eta \cdot \nabla_\theta \mathcal{L}(\theta)$;
**end**

---

**Algorithm 2:** Testing Stage Sampling with History, Policy and Realism Guidance

---

**Input:** History conditioned diffusion model $\mathbf{D}_\theta$, victim's policy $\pi$, number of denoising steps $T$, autoencoder-based unrealism detector $\mathbf{AE}$, agent state-action value function $Q^\pi$, given **true** history $\tau_{t-1} = \{s_{t-k}, a_{t-k}, \ldots, s_{t-1}, a_{t-1}\}$ for SHIFT-O or **observed** history $\tau_{t-1} = \{o_{t-k}, a_{t-k}, \ldots, o_{t-1}, \varnothing\}$ for SHIFT-I, classifier-free guidance strength $\Gamma_1$, classifier guidance strength $\Gamma_2$, attack rate $\xi$, true state $s_t$, history length $k$.

**Output:** Generated sample $\tilde{s}_t$

---

**Initialize:** $\tilde{s}_t^T \sim \mathcal{N}(0, \mathbf{I})$;

Calculate state importance $\omega(s) = \max_{a_1 \in A} Q^\pi(s, a_1) - \min_{a_2 \in A} Q^\pi(s, a_2)$;

**if** $\omega(s_t)$ *in top $\xi$ percentile of previous states and total attacked times $< t \times \xi$* **then**

    *total attack times* += 1;

    // Inject SHIFT Attack

    **for** $i = T$ **to** 1 **do**

        // Calculate proposed output $\hat{s}_t$ based on $\tilde{s}_t^i$

        $\hat{s}_t = \tilde{s}_t^i$ ;

        **for** $j = i$ **to** 1 **do**

            $\hat{s}_t = \mathbf{D}_\theta(\hat{s}_t, c_{\text{noise}}^j, \tau_{t-1})$

        **end**

        Calculate policy guidance gradient $g \leftarrow \nabla_{\hat{s}_t} Q^\pi(s_t, \pi(\hat{s}_t))$;

        Inject policy guidance $\tilde{s}_t^i = \tilde{s}_t^i - \Gamma_2 g$;

        Generate next sample $\tilde{s}_t^{i-1} = \Gamma_1(i)\mathbf{D}_\theta(\tilde{s}_t^i, c_{\text{noise}}^i, \tau_{t-1}) + (1 - \Gamma_1(i))\mathbf{D}_\theta(\tilde{s}_t^i, c_{\text{noise}}^i)$;

        **if** $i \neq 1$ **then**

            Conduct a gradient descent based on the reconstruction error from the unrealism detector

$$\tilde{s}_t^{i-1} = \tilde{s}_t^{i-1} - \nabla_{\tilde{s}_t^{i-1}} \mathcal{L}(\tilde{s}_t^{i-1}, \mathbf{AE}(\tilde{s}_t^{i-1}));$$

        **end**

    **end**

**end**

---

## C.7 Hyper-parameters Setting

**EDM Diffusion Model Training Parameters.** As mentioned before, we only change the residual block layers from [2,2,2,2] to [2,2] and the denosing steps to 5, and set the drop condition rate to 0.1. We keep other hyperparameters the same as Alonso et al. [3] for training the EDM diffusion model.

**Testing Stage Parameters and Testbench Specification.** We schedule the classifier-free guidance scale as $\Gamma_1(i) = \max(\frac{T-i}{T}, 0.3)$, where $T$ is the number of reverse steps and $i$ is the current reverse step. We set the policy guidance strength $\Gamma_2$ differently in each environment under each defense. In the Pong environment, we set $\Gamma_2 = 3.5$ for DQN, DP-DQN, and DMBP and $\Gamma_2 = 2$ for all other defenses. In the Freeway environment, we set $\Gamma_2 = 6$ for DQN, DP-DQN and DMBP and $\Gamma_2 = 4.5$ for all other defenses. In the BankHeist environment, we set $\Gamma_2 = 4$ for all defenses. In the RoadRunner environment, we set $\Gamma_2 = 6$ for all defenses. In the Doom environment, we set $\Gamma_2 = 4.5$ for DQN, DP-DQN, and DMBP and $\Gamma_2 = 2.5$ for all other defenses. For the AirSim autonomous driving simulator, we utilize CITY, a complex environment that simulates real-world city traffic situations. We set $\Gamma_2 = 0.5$ for all defenses in AirSim.

We conduct all our experiments on a workstation equipped with an Intel I9-12900KF CPU, an RTX 3090 GPU, and 64GB system RAM.

**Pre-processing Atari, Doom and AirSim Environments.** We have used the same environment wrappers as in Zhang et al. [60], which convert an RGB image to a gray-scale image and resize the image to reduce its resolution from $210 \times 160$ to $84 \times 84$ for Atari environments and $320 \times 240$ to $84 \times 84$ for Doom. We also follow Zhang et al. [60] to center crop images using the same shifting parameters as in Zhang et al. [60], where we set the cropping shift to 10 for Pong, 20 for Roadrunner, and 0 for Freeway, Bankhesit, and Doom. We also convert the front view camera snapshot of the autonomous agent in AirSim to $84 \times 84$ grayscale image. We do not stack frames in our pre-processing.

# D More Evaluation Results

Table 3: Attack results on Atari BankHeist and RoadRunner Environments under SHIFT-I and SHIFT-O attacks with 1.0 and 0.25 attack frequency.

| | BankHeist | | | | RoadRunner | | | |
|---|---|---|---|---|---|---|---|---|
| Model | SHIFT-O-1.0 | SHIFT-O-0.25 | SHIFT-I-1.0 | SHIFT-I-0.25 | SHIFT-O-1.0 | SHIFT-O-0.25 | SHIFT-I-1.0 | SHIFT-I-0.25 |
| DQN-No Attack | 680.0±0.0 | 680.0±0.0 | 680.0±0.0 | 680.0±0.0 | 13500.0±0 | 13500.0±0 | 13500.0±0 | 13500.0±0 |
| DQN | 0.0±0.0 | 50.0±45.6 | 0.0±0.0 | 0.0±0.0 | 0.0±0.0 | 160.0±151.7 | 0.0±0.0 | 0.0±0.0 |
| SA-DQN | 3.0±3.5 | 80.0±14.1 | 0.0±0.0 | 0.0±0.0 | 480.0±521.5 | 2820.0±1690.3 | 0.0±0.0 | 440.0±559.5 |
| WocaR-DQN | 6.0±5.5 | 66.0±27.0 | 0.0±0.0 | 0.0±0.0 | 120.0±83.7 | 1080.0±807.5 | 0.0±0.0 | 100.0±70.7 |
| CAR-DQN | 12.0±14.1 | 24.0±15.2 | 0.0±0.0 | 6.0±8.9 | 1100.0±447.2 | 920.0±44.7 | 260.0±313.0 | 540.0±320.9 |
| DP-DQN | 15.0±12.0 | 32.0±29.2 | 0.0±0.0 | 0.0±0.0 | 220.0±130.4 | 3900.0±2640.1 | 0.0±0.0 | 0.0±0.0 |
| DMBP | 20.0±18.2 | 54.0±53.7 | 0.0±0.0 | 0.0±0.0 | 240.0±134.2 | 4300.0±1433.5 | 0.0±0.0 | 80.0±83.7 |

Table 4: Attack results of our methods with {0.15,0.25,0.5,1.0} attack frequencies against DMBP defenses.

| | | SHIFT-O | | | | SHIFT-I | | | |
|---|---|---|---|---|---|---|---|---|---|
| Frequency | No Attack | 0.15 | 0.25 | 0.50 | 1.00 | 0.15 | 0.25 | 0.50 | 1.00 |
| Pong | 21.0±0.0 | -0.6±3.0 | -9.8±4.5 | -12.6±3.8 | -14.0±4.9 | -20.4±1.3 | -20.2±0.8 | -20.6±0.9 | -20.6±0.9 |
| Freeway | 34.1±0.1 | 31.6±0.5 | 31.0±1.2 | 28.8±1.5 | 22.0±0.7 | 13.8±2.2 | 14.6±2.5 | 14.6±1.1 | 19.2±2.9 |
| Doom | 75.4±4.4 | 71.4±5.1 | 65.4±9.8 | 65.2±7.1 | -184.8±231.7 | -101.6±200.3 | -256.2±180.4 | -239.8±174.4 | -302.0±2.7 |
| AirSim | 40.3±0.5 | 28.1±7.2 | 22.8±8.0 | 19.2±6.6 | 20.8±12.0 | 11.7±3.5 | 9.4±4.9 | 6.6±2.5 | 6.2±0.7 |

Table 5: Attack performance on the Highway environment (with a continuous action space) and the Pong environments (with a discrete action space), under PPO policies with/without DMBP defense. For the continuous action environment Highway, we compare with the MAD attack from Zhang et al. [60]. For the discrete action environment Pong, we compare with the PGD attack.

| | Highway | | Pong | |
|---|---|---|---|---|
| Attack Type | PPO | DMBP | PPO | DMBP |
| No Attack | $23.3 \pm 9.6$ | $22.5 \pm 8.9$ | $21.0 \pm 0.0$ | $21.0 \pm 0.0$ |
| MAD/PGD Attack | $2.7 \pm 1.3$ | $16.8 \pm 3.4$ | $-21.0 \pm 0.0$ | $20.8 \pm 0.4$ |
| SHIFT-O-1.0 | $2.0 \pm 1.8$ | $4.4 \pm 0.6$ | $-21.0 \pm 0.0$ | $-12.3 \pm 7.6$ |
| SHIFT-O-0.25 | $13.7 \pm 6.9$ | $15.4 \pm 5.2$ | $-20.6 \pm 0.9$ | $1.6 \pm 4.2$ |
| SHIFT-I-1.0 | $0.84 \pm 0.3$ | $2.54 \pm 0.2$ | $-21.0 \pm 0.0$ | $-20.0 \pm 1.2$ |
| SHIFT-I-0.25 | $2.55 \pm 0.8$ | $3.80 \pm 1.2$ | $-21.0 \pm 0.0$ | $-19.6 \pm 0.8$ |

Table 6: Automated detection results with MAD threshold set to 5, and CUSUM configured with a drift of 1.5 and a threshold of 3.

| Env | Freeway | |
|---|---|---|
| Attack Method | MAD Dector | CUSUM Dector |
| PGD-1/255 | Undetected | Undetected |
| PGD-15/255 | Detected | Detected |
| MinBest-1/255 | Detected | Detected |
| MinBest-15/255 | Detected | Detected |
| PA-AD-1/255 | Undetected | Undetected |
| PA-AD-15/255 | Detected | Detected |
| PA-AD-TC-15/255 | Detected | Detected |
| Rotation Degree 1 | Undetected | Undetected |
| Transform(1,0) | Undetected | Undetected |
| SHIFT-O-1.0 | **Undetected** | **Undetected** |
| SHIFT-I-1.0 | **Undetected** | **Undetected** |

## D.1 Additional Attack Results in Atari Environments

We report additional attack performance results on the remaining two Atari environments: BankHeist and RoadRunner in Table 3.

## D.2 Attack Performance on Continuous Action Space Environment and PPO Policy

Table 5 shows our SHIFT attack can successfully work on PPO policies under both continuous action space environment Highway [34] environment and discrete action space environment Atari Pong.

Table 7: DIRE [49] Detection Results

| Env | SHIFT-O | SHIFT-I | Clean Diffusion |
|---|---|---|---|
| **Pong** | 3.19% | 3.56% | 97.63% |
| **Freeway** | 4.15% | 4.72% | 95.46% |

We compared our attack with the Maximal Action Difference (MAD) attack used in [60], which is a simple yet effective attack against the PPO algorithm, both with and without the DMBP defense. Since MAD attack does not apply on discrete action space, we report PGD attack with a budget $1/255$ as a baseline. The results demonstrate that our attack outperforms MAD/PGD in both cases, further supporting its generalizability to different RL algorithms.

### D.3 Detection Results

Furthermore, inspired by the adversarial attack detection method in Russo and Proutiere [42], we have evaluated our attack under two commonly used anomaly detection methods based on the Wasserstein-1 distance between adjacent perturbed states: (1) the MAD (Median Absolute Deviation) anomaly detector [1], which flags anomaly if three consecutive steps of perturbed trajectories violate clean trajectories' median + threshold$\times$MAD, where clean trajectories' median and MAD are both evaluated under Wasserstein-1 distance between adjacent true states. We set threshold = 5 in the table below; and (2) CUSUM [41], a sequential change detection method that identifies persistent deviations (again in terms of Wasserstein-1 distance) from clean trajectories by accumulating small shifts over time. We set the drift to 1.5 and threshold to 3 in CUSUM. We have compared PGD, Minbest, PA-AD attacks (with budgets $1/255$ and $15/255$), Rotation and Transformation attacks from Korkmaz [31] and our method.

Furthermore, multiple methods have been proposed to detect whether an image is original or generated by a diffusion model. The victim can apply these detection methods to determine whether the observed state is diffusion-generated. One of the most widely used approaches is DIRE [49], which detects diffusion-generated images by examining their reconstruction behavior through a pretrained diffusion model. Given an input image $x_0$, DIRE first adds Gaussian noise to obtain $x_t$ at diffusion step $t$, and then denoises it using the model's reverse process to reconstruct $\hat{x}_0$. The reconstruction error $E_t = |x_0 - \hat{x}_0|$ serves as a key indicator of authenticity.

In DIRE, *real* images typically produce **larger** reconstruction errors, whereas diffusion-generated images align more closely with the model's output distribution and thus yield **smaller** errors. Leveraging this separation, DIRE classifies inputs without retraining and generalizes across architectures such as ADM, GLIDE, and Stable Diffusion.

We conduct an initial investigation of DIRE against our SHIFT attack (Table 7).We train a binary classifier with clean images' DIRE reconstruction features and normal diffusion generated images' DIRE reconstruction features. The classifier successfully flags normal diffusion outputs in the no-attack setting. However, it fails to detect both SHIFT variants, labeling them as normal states. We conjecture that SHIFT's policy guidance method pushes generated states away from the diffusion model's manifold in a manner that *increases* reconstruction error, thereby imitating the high-error signature of real images and evading DIRE's decision rule.

### D.4 Ablation on Attack Frequency

Table 4 illustrates the performance of our attack across different attack frequencies $\xi$. The results show that even at low attack frequencies, our method significantly reduces the agent's cumulative reward. Moreover, the reduction becomes more pronounced as the attack frequency increases.

### D.5 Ablation on DDPM and EDM diffusion architectures

We compare DDPM and EDM in terms of attack efficiency and computational cost in Table 11. The results show that EDM and DDPM exhibit similar attack performance. However, DDPM is significantly slower than EDM in terms of sampling time (the average time needed to generate a single perturbed state during testing), making DDPM incapable of generating real-time attacks during

Table 8: Performance of high-sensitivity direction attacks in [31].

| | | Pong | Freeway |
|---|---|---|---|
| Attack | Defense | Reward↓ | Reward↓ |
| B&C | DQN | $-21 \pm 0.00$ | $23 \pm 0.00$ |
| | SA-DQN | $11 \pm 0.00$ | $25 \pm 0.00$ |
| | DMBP | $20 \pm 1.41$ | $27.2 \pm 0.68$ |
| Blurred Observations | DQN | $-21 \pm 0.00$ | $18 \pm 0.00$ |
| | SA-DQN | $-20 \pm 0.00$ | $27 \pm 0.00$ |
| | DMBP | $20 \pm 0.58$ | $33.2 \pm 0.37$ |
| Rotation Degree 1 | DQN | $-20 \pm 0.00$ | $26.6 \pm 0.45$ |
| | SA-DQN | $-18 \pm 0.00$ | $21 \pm 0.00$ |
| | DMBP | $14.6 \pm 2.68$ | $27.6 \pm 0.45$ |
| Transform (1,0) | DQN | $-21 \pm 0.00$ | $26 \pm 0.00$ |
| | SA-DQN | $-21 \pm 0.00$ | $24 \pm 0.00$ |
| | DMBP | $17.8 \pm 2.85$ | $27.2 \pm 0.37$ |

Table 9: Large-scale rotation and shift attacks against fine-tuned diffusion-based defense.

| Pong | Defense | Reward↓ |
|---|---|---|
| Rotation Degree 3 | DMBP | $20 \pm 0.71$ |
| Transform (2,1) | DMBP | $18.8 \pm 1.79$ |

Table 10: Average Reconstruction error, Wasserstein distance, SSIM and LPIPS of the high-sensitivity direction attacks in Korkmaz [31] across a randomly sampled episode. The Wasserstein distance is the Wasserstein-1 distance calculated between the current perturbed state and the previous step's true state.

| Freeway | Reconstruction Error↓ | Wass.($\times 10^{-3}$)↓ | SSIM↑ | LPIPS↓ |
|---|---|---|---|---|
| B&C | $15.30 \pm 0.02$ | $0.80 \pm 0.2$ | $0.9429 \pm 0.0006$ | $0.0455 \pm 0.0031$ |
| Blurred Observations | $5.81 \pm 0.04$ | $0.82 \pm 0.2$ | $0.5974 \pm 0.0034$ | $0.3535 \pm 0.0202$ |
| Rotation Degree 1 | $6.41 \pm 0.2$ | $0.80 \pm 0.2$ | $0.8237 \pm 0.0022$ | $0.0351 \pm 0.0041$ |
| Transform (1,0) | $9.32 \pm 0.1$ | $0.83 \pm 0.2$ | $0.6045 \pm 0.0074$ | $0.0741 \pm 0.0068$ |
| SHIFT-O-1.0 | $1.02 \pm 0.5$ | $0.89 \pm 0.3$ | $0.9990 \pm 0.0014$ | $0.0008 \pm 0.0014$ |
| SHIFT-I-1.0 | $1.05 \pm 0.5$ | $0.84 \pm 0.2$ | $0.9904 \pm 0.0043$ | $0.0175 \pm 0.0124$ |

Table 11: Ablation studies on EDM and DDPM diffusion architectures.

| Pong | DDPM | | EDM | |
|---|---|---|---|---|
| | Reward ↓ | Deviation Rate(%)↑ | Reward↓ | Deviation Rate(%)↑ |
| DQN | $-20.6 \pm 0.5$ | $83.6 \pm 1.0$ | $-21.0 \pm 0.0$ | $90.0 \pm 0.3$ |
| DMBP | $-10.4 \pm 4.8$ | $46.1 \pm 0.3$ | $-14.0 \pm 6.2$ | $47.3 \pm 0.9$ |
| Sampling Time | $\sim 5$ sec | | $\sim 0.2$ sec | |

testing. This validates the selection of EDM as the diffusion model architecture for constructing our attacks.

## D.6 Performance and Stealthiness of High-Sensitivity Direction Attacks in Korkmaz [31]

Korkmaz [31] proposes various high-sensitivity direction-based attacks that can generate perturbed states that are visually imperceptible and semantically different from the clean states, including changing brightness and contrast (B&C), image blurring, image rotation and image transform. These attack methods reveal the brittleness of robust RL methods such as SA-DQN, but they mainly target changes in visually significant but not domain-specific semantics. For example, the relative distance between the pong ball and the pad will remain the same after brightness and contrast changes or image transform in the Pong environment. Consequently, the perturbed images generated by these

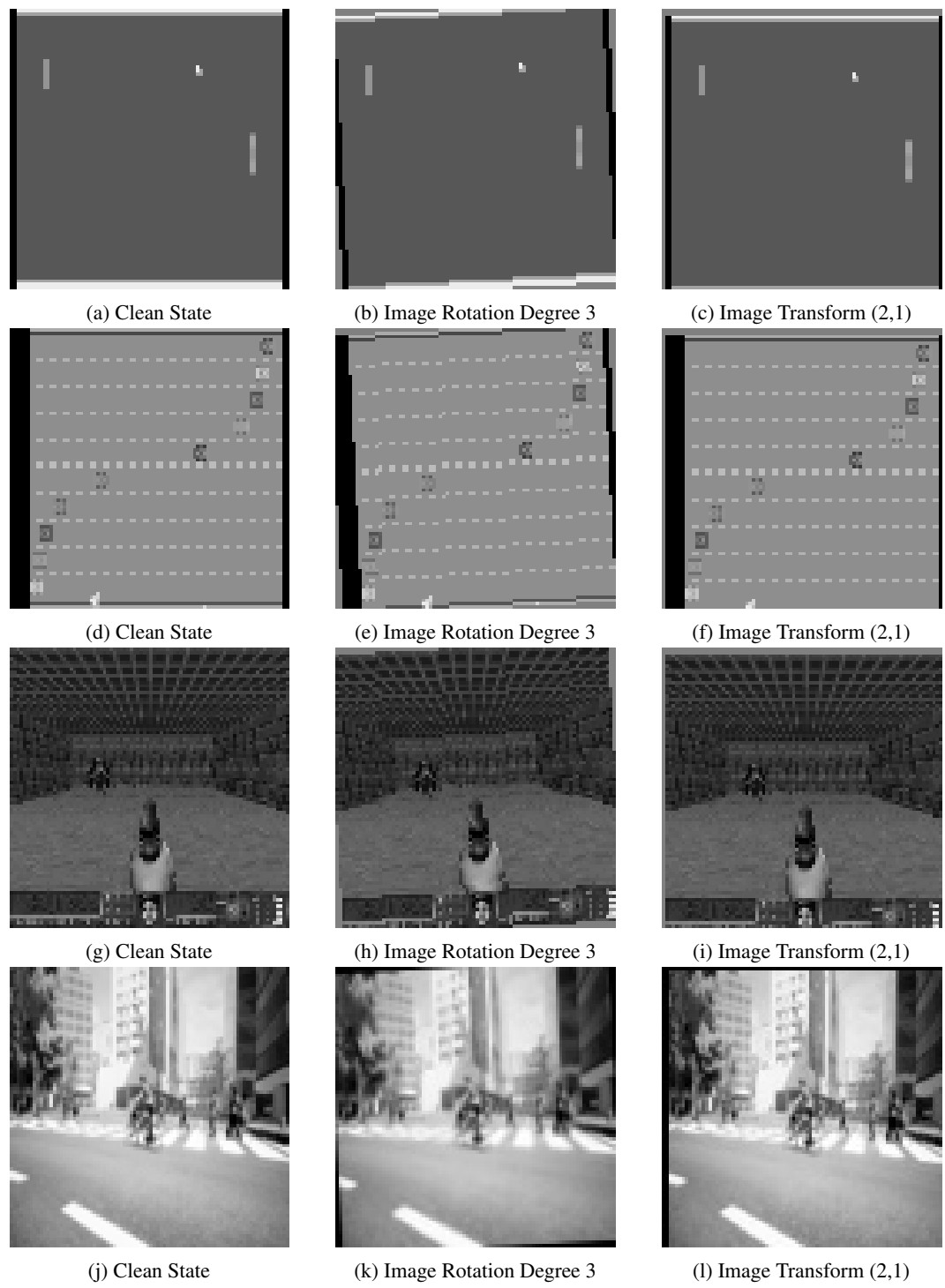

(a) Clean State      (b) Image Rotation Degree 3      (c) Image Transform (2,1)

(d) Clean State      (e) Image Rotation Degree 3      (f) Image Transform (2,1)

(g) Clean State      (h) Image Rotation Degree 3      (i) Image Transform (2,1)

(j) Clean State      (k) Image Rotation Degree 3      (l) Image Transform (2,1)

Figure 7: Perceptual impact of large-scale rotation and transform on Atari Pong, Atari Freeway, Doom, and Airsim.

methods can potentially be purified by a diffusion model. To confirm this, we have conducted new experiments, showing that (1) the DMBP defense with a diffusion model trained from clean data only is able to defend against B&C, blurring, and small scale rotation and transform attacks (see Table 8), and (2) when the diffusion model is fine-tuned by randomly applying image rotations or transform during training, the DMBP defense can mitigate large scale image rotations and transform

considered in Korkmaz [31] (see Table 9). In contrast, our diffusion guided attack can change the decision-relevant semantics of the images, such as moving the Pong ball to a different position without changing other elements in the Pong environment as shown in Figure 2). This is the key reason why our attack can bypass strong diffusion-based defense methods.

Furthermore, Korkmaz [31] claims their attacks are imperceptible by comparing the perturbed state $\tilde{s}_t$ and the true state $s_t$. However, we found that this only holds for small perturbations. For example, the Rotation attack with degree 3 and Transform attack (1,2) in the Pong environment considered in their paper can be easily detected by humans (see Figure 7). Further, their metric for stealthiness is static and does not consider the sequential decision-making nature of RL. In contrast, our attack method aims to stay close to the set of true states $S^*$ to maintain static stealthiness and realistic (Definitions 3.2) and align with the history to achieve dynamic stealthiness (Definitions 3.4). These are novel definitions for characterizing stealthiness in the RL context. The static stealthiness and realism are demonstrated through the low reconstruction loss of our method, shown in Table 2,10. In contrast, attacks from Korkmaz [31] generate out-of-distribution perturbed states by applying image transformations, which induce high reconstruction error, indicating that they are less realistic than our methods. We further compare the Wasserstein distance between a perturbed state and the previous step's perturbed state, SSIM, and LPIPS metrics to measure the historical alignment and trajectory faithfulness in Table 10. The results show that the perturbed states generated by Korkmaz [31] achieve the same level of historical alignment but are less trajectory-faithful than our SHIFT attack.

## D.7 Time Complexity Comparison

In terms of time complexity, high-sensitivity direction attacks can generate perturbations instantaneously as they are policy independent, and PGD, PA-AD-TC, MinBest, and PA-AD all take around 0.02 seconds to generate a perturbation with 10 iterations. Due to the computational overhead of the reverse process, diffusion-based methods typically require longer generation times. However, by adopting the EDM diffusion paradigm, we reduce the reverse process steps to 5, resulting in a generation time of approximately 0.2 seconds per perturbed state. Although slower than PGD, MinBest, and PA-AD, this still allows our attack to remain feasible for real-time applications.

