# OpenReview forum: "Diffusion Guided Adversarial State Perturbations in Reinforcement Learning"
_NeurIPS.cc/2025/Conference — NeurIPS 2025 poster_

### Official Review · Reviewer_3Yfq · 2025-06-27

**Clarity:** 1
**Significance:** 2
**Originality:** 2
**Rating:** 4
**Confidence:** 3

**Summary:**

This paper proposes SHIFT, a diffusion-based state perturbation adversarial attack against RL agents, which focuses on creating perturbed states that are (i) semantically different, (ii) history-aligned, and (iii) trajectory-faithful. Two variants of SHIFT are proposed, SHIFT-O, conditioned on the true trajectory of the victim agent, and SHIFT-I, conditioned on the victim agent’s observed trajectory (i.e., with states subject to SHIFT perturbations). The proposed attack is motivated by shortcomings of existing attacks and is evaluated on multiple environments and considering different existing defense approaches. The main evaluation considers attack effectiveness, attack realism, historical alignment, and trajectory faithfulness. Additional results regarding computational complexity, ablations, and tradeoffs are included in the appendix.

**Questions:**

Regarding C1:
Q1a- How are lines 391-392 backed by the numerical results exactly? And most importantly, what is the precise notion of undetectability/stealthiness considered by the authors (human, automated, both)?
Q1b- The ablation on realism guidance in Figure 6a does not seem meaningful, is it not expected that the reconstruction error will decrease if SHIFT does actively minimize it?

Regarding C2:
Q2- How well can diffusion-generated image detectors such as DIRE [B] detect perturbed states generated by SHIFT?

Regarding C3:
Q3- The formulation in [59] is not exclusive to attacks bounded by an L_p norm. Why was its considered perturbation set not adapted to use, e.g., the reconstruction error from the auto-encoder, or some other notion better aligned with SHIFT?

Regarding C4:
Q4a- How does the trilemma follow directly from the definition of the SHIFT framework?
Q4b- How does Table 2 present conclusive evidence that no other method could dominate the ones shown in all key metrics? Also, wouldn’t it be possible for a method to dominate all others while still having to tradeoff said key metrics? Can anything be said about how any of these methods can tradeoff one key metric for another?
Q4c- The statement that only two out of three properties can be optimized at once (lines 333-334 and 408-410) is not clear. Does this refer to two specific properties, if so, which ones? Alternatively, does it refer to any possible pair of properties? How is this claim supported by the SHIFT framework or the numerical results?

Regarding C5:
Q5a- Why were the Freeway environment and the DP-DQN defense chosen for the “key metric” evaluations?  How do these key metrics look like with different defenses or when considering only the vanilla DQN? Are there any relevant differences when considering other environments?

Regarding C6:
Q6a- What architecture and training scheme were used for the autoencoder? The policy guidance (line 303) seems to perform a single gradient descent update with a step size of one, why was this chosen over multiple steps and/or a smaller step size? What would be the expected impact of modifying this policy guidance step?
Q6b- What sort of resources are considered for the “20 hours” needed to train GRAD and PROTECTED? How do they compare to the resources used by the authors?

Additional questions:
Q7- While the threat model is not far away from prior work on RL (e.g., [16]), it is quite different from prior work on computer vision. What is the main application area of the current work? In what setting can one assume the attacker is able to completely alter the visual input of the victim? What is its practical relevance? In line 104 the authors refer to this as a “worst-case scenario”, but this is worst-case from the perspective of the victim, how can one justify the relevance of an attack in this setting, which is a best-case scenario for the attacker?

Q8- In line 293 the authors state “Thus s_t (perturbed) is semantically distinct from the true state s_t and achieve attack performance”. What is meant by “attack performance”? How does semantic distinction from the true state entail attack effectiveness? Is it not the case that the attacks that allegedly fail to cause a semantic change can also be effective (while some attacks that cause a semantic change can be ineffective)?

Q9- What is meant by “which agent has no access during testing” in line 374?

Q10- What is the difference between D(.) and Proj_{S*}(.)? Why is it necessary to introduce D(.) in line 186? Why is “(after projection onto S*)” included in definitions 3.4 and 3.5? Is it not implied that all observed states are perturbed in this scenario and therefore already projected onto S*?

Q11- The authors mention a well-trained policy “with exploration to ensure coverage” in line 243. What does this mean exactly?

Q12- What is the meaning of “connects the generative models and optimization methods” in line 849?

**Ethical Concerns:**

["NO or VERY MINOR ethics concerns only"]

**Final Justification:**

Authors have address most of my comments during the rebuttal.

**Limitations:**

While the authors include an Impact Statement in the appendix, the limitations of the work are mainly reduced to stating that an attack that “better satisfies” the three properties that SHIFT aims to achieve remains an open problem. Limitations relating to the threat model assumptions (see Q7) or potential defenses and notions of stealthiness that are not explored (see C2 and C3) may require further discussion.

**Quality:**

2

**Strengths And Weaknesses:**

Strengths:
S1- Intuitive description and illustrations for SHIFT-O and SHIFT-I, which highlight how it is distinct from existing attacks
S2- Concrete definitions for the three fundamental attack properties of SHIFT
S3- Numerical results comparing SHIFT to existing attacks and evaluating its effectiveness against existing defenses
S4- Additional ablations and results in the appendix provide a more comprehensive evaluation of SHIFT
S5- Code is provided to ensure reproducibility

Weaknesses:
W1- Inconsistent motivation: in particular, the fact that SHIFT aims for semantic changes is counterintuitive. In line 54 the authors argue that the attacks from [31] are not stealthy towards humans, however, when considering prior works on adversarial attacks in general, retaining semantic consistency between the perturbed and original image would make more sense if the goal were stealthiness towards humans. For example, this has been argued by [A]. In short, even though some experimental results aim to demonstrate SHIFT’s stealthiness, it is quite counterintuitive that an attack encouraging semantic changes should be expected to be stealthy.

W2- Unconvincing evaluation: various aspects of the numerical results raise questions. For example, using the auto-encoder reconstruction error to assess realism seems unfair, since unlike other methods, SHIFT is optimized to be consistent with the auto-encoder’s notion of realism. Figure 6c seems more compelling (although l2 distance to true states is still not an ideal metric for realism), yet this figure is never explained in-text (not even in the appendix) which makes it hard to interpret, what does the “conditional diffusion model” correspond to in that plot? The choice of environments, baseline attacks, and defenses also raises some questions (see “Major Comments” below).

W3- Issues with writing/structure: the paper contains many typos and the writing is often unclear, hurting the clarity of the paper (some specific instances are listed below). In terms of structure, the way content corresponding to related work and results is split between the main paper and the appendix makes it hard to understand SHIFT’s motivation and advantages.

Reference:

[A] Hosseini, Hossein and Radha Poovendran. Semantic Adversarial Examples. In Proc. of Conference on Computer Vision and Pattern Recognition Workshops (CVPRW), 2018.

Major Comments

C1- Stealthiness/undetectability is introduced as one of the main features of SHIFT (line 13), yet its evaluation is not convincing. Lines 90-91 claim that the stealthiness of SHIFT is demonstrated by the reconstruction error, Wasserstein-1 distance, LPIPS and SSIM. As mentioned in W2, the reconstruction error is not a fair metric since SHIFT actively minimizes it, unlike other methods. In lines 391-392 the authors claim they outperform all baselines in terms of stealthiness, but Table 2 shows many baselines are better in terms of Wasserstein-1 distance, and for SHIFT-I this is also the case regarding SSIM and LPIPS. A more realistic assessment of the performance follows in lines 393-399, but the opening statement in lines 391-392 is quite confusing. Moreover, the results only cover the Freeway environment with the DP-DQN defense, which further weakens this claim even if SHIFT-I and SHIFT-O indeed dominated all baselines, so it would be helpful to justify why this is the only scenario used to compare to other methods.

C2- Table 6 in the appendix shows some results on detectability using a MAD and a CUSUM detector, however these detectors and their parameters are not justified. Providing the scores from the detectors instead of their thresholded binary results would be more insightful. Additionally, it would be meaningful to discuss methods that detect diffusion-generated images such as DIRE [B].
[B] Wang et al. DIRE for Diffusion-Generated Image Detection. In Proc. of International Conference on Computer Vision (ICCV), 2023.

C3- Regarding the evaluated defenses, while SHIFT is effective in bypassing defenses against bounded L_p norm attacks, which use adversarial training or diffusion, the fact that they assume traditional attacks (with a bound on their L_p norm) makes them unsuitable to properly evaluate SHIFT’s effectiveness against a defended model. Also, regarding the diffusion-based defenses, even though they are based on diffusion they do not assume diffusion-based attacks, so in this sense an evaluation using DIRE [B] or other defenses against diffusion-based perturbations would be more meaningful. Even for the appendix results in Table 8 regarding [31], the DMBP defense was finetuned to better handle the attacks proposed by [31], and while the authors explain why SHIFT works against diffusion-based defenses in lines 1050-1053, the finetuning of DMBP for [31] raises the question of why no defenses were adjusted to work better against SHIFT.

C4- The “fundamental trilemma” from lines 332-335 is unclear. It is contradictory to the fact that SHIFT aims to optimize the three properties at once. Also, the later statement in lines 399-400 is not supported by the numerical results (and is in direct contradiction with lines 391-392, which suggest that SHIFT achieves stronger attack performance and better stealthiness). Additionally, each component in this “trilemma” corresponds to more than one key metric, further complicating the interpretation of lines 399-400.

C5- Only the scenario with the Freeway environment and the DP-DQN defense is used for the “key metric” evaluations. Even in the appendix these results are omitted for all other defenses/environments. This seems unfair since DP-DQN is a defense against norm-bounded attacks (like most of the baseline attacks), thus evaluating the key metrics only when such a defense is present puts into question the supposedly superior performance of SHIFT.

C6- Important implementation details are missing. Although the authors provide code in the supplementary material, the training and architecture of the realism enhancement autoencoder is not described with sufficient detail in the paper. Additionally, the authors mention GRAD and PROTECTED “take more than 20 hours” to train (lines 813-815), but it is unclear how does this compare to SHIFT or other approaches, since only test-time cost is reported for SHIFT.

C7- As a final comment, the discussion of numerical results seems somewhat shallow for various results. For example, in Sections 4.1 and D.1 there is no discussion regarding differences (or similarities) between the different environments, similarly in D.6 the authors mention “the reduction becomes more pronounced as the attack frequency increases” in line 1029, but do not comment on certain outliers like SHIFT-I in Freeway (reward increases with frequency) or the high variance for multiple scenarios in the Doom environment.

Other typos/unclear passages:
1- “environments with raw pixel images as input” suggests that the environments (and not the agents) have image inputs.
2- “either can…” in line 53.
3- It is unclear what the authors mean by “stealthy from human perspectives” in line 54.
4- The meaning of “classifier-free guidance” (line 80) is not explained when introduced.
5- In lines 87 and 91, authors use “and” instead of commas when listing different elements, which can be confusing. This is also the case in lines 224-227.
6- In line 118 the statement of “being detected” does not specify who/what is assumed to perform the detection.
7- In (1) it seems “t” should be “i". It is also unclear why the theta that parametrizes epsilon also parametrizes mu and sigma, especially as line 136 states that sigma can be “predicted through a neural network or set by a predetermined scheduler”. Moreover, theta is not properly introduced.
8- “realistic perturb states” in line 145.
9- “quantified metrics” in line 216.
10- “the training the” in line 230.
11- “the static and dynamic views” in line 235 has an unclear meaning (see also “static and dynamic perspectives” in lines 734-735).
12- “the Freeway” in line 258.
13- “under true state” and “victim’s performance loss” in lines 277-278.
14- “achieve attack performance” in line 293.
15- “Figures 2” in line 319.
16- Lines 333-335 are quite unclear, see also C4 above and Q4a, Q4b, and Q4c below.
17- “any attack then applies” in line 351, “Algorithm 1,2” in line 353.
18- The environments used for evaluation are not clearly introduced, line 355 refers to “four Atari environments, Doom game, and Airsim autonomous driving simulator”, yet Table 1 presents results for Pong, Freeway, Doom, and Airsim, and Table 2 focuses on Freeway. Section D.1. in the appendix refers to “the remaining two Atari environments: BankHeist and Roadrunner”, clarifying that Pong and Freeway are the other two Atari environments, but this is not clear when introduced in Section 4. Table 5 also refers to a “Highway” environment. Additionally, it is redundant to include the environment for tables that present results for a single environment (Tables 2, 6, 8, 9, and 10), and sometimes “Env”/”Environment” or environment names are included erroneously (see Tables 4, 8, 9, and 10).
19- “which agent has no access during testing” in line 374.
20- “indicating less trajectory faithful” in line 398.
21- The expression in line 773 would be more clearly explained with words.
22-“decreases the searching space” in line 779.
23-“in solving” in line 839.
24- “traditional _p-norm” in line 877.
25-“function” in line 934.
26- “random” in Algorithm 1 is not properly defined.
27- “that simulate real city traffic” in line 978.
28- the Y label “Percentage (0-1)” in Figure 6b is unclear, and it is hard to see how it relates to the deviation rate mentioned in the figure caption.
29- “table below” in line 1006 seems to refer to Table 6, but this is not obvious.
30-“transform” in line 1049.
31-“realistic” in line 1059, “Definitions 3.2” and “Definitions 3.4” in line 1060. “Table 2,9” in line 1062.
32-“indicating less realistic than our methods” in line 1064

---

> ### Author Rebuttal · Authors · 2025-07-31
>
> Dear Reviewer 3Yfq:
>
> Thanks for your insightful feedback and we will address your concerns.
>
> Q1a: Semantic change is not considered as stealthy. What is the exact definition of stealthiness?
>
> A1a: In the state perturbation threat model, the agent only observes perturbed states and has no access to the true states, unlike in traditional vision settings. Therefore, semantic changes can be considered stealthy if they remain physically realistic and plausible, as defined in Definition 3.2.
>
> Our definition of stealthiness is grounded in three apsectives: Realism (Def. 3.2), Historical Alignment (Def. 3.4), and Trajectory Faithfulness (Def. 3.5). Due to the computational intractability of projection onto the valid states, we use approximate metrics to evaluate stealthiness: autoencoder reconstruction error (for realism), Wasserstein distance (for historical alignment), and SSIM/LPIPS (for trajectory faithfulness).
>
> Table 2 provides quantitative evidence: SHIFT achieves strong attack performance while maintaining stealth, as indicated by lower reconstruction error and Wasserstein distance comparable to low-budget baselines. Additional detection results in Appendix D and preliminary DIRE outcomes (see Q2) reinforce this finding. Although SSIM and LPIPS are not directly usable for detection—since they require access to the true states—they help assess how trajectory faithful the perturbations are.
>
> In summary, SHIFT surpasses all other baselines in attack performance and has comparable stealthiness with low-budget baselines, justifying our claim in lines 391–392.
>
> Q1b: Unfair comparison for reconstruction error.
>
> A1b: The unfair comparison concern is addressed by our ablation study (Figure 6a): SHIFT-O without autoencoder-based realism enhancement achieves reconstruction error of 1.06±0.5 in Freeway and is still lower than all baseline attacks (In Table 2). This occurs because our conditional diffusion model is trained upon clean trajectories and can generate realistic states. The conditional diffusion model ensures physical plausibility, making SHIFT's realism advantage intrinsic rather than evaluation-biased.
>
> Q2: Diffusion generated sample detector.
>
> A2: We conducted a preliminary evaluation of the DIRE detector. Given an input image $x$, DIRE first performs a reverse diffusion sampling process to add noise until reaching the standard normal prior $\mathcal{N}(0, 1)$, then generates a new image $x’$ via the diffusion sampling process. The DIRE distance is defined as the $l_2$ distance between $x$ and $x’$. Since diffusion-generated images tend to remain closer to the diffusion sampling distribution and the real image distribution is different from the diffusion sampling distribution, DIRE assumes that real images will have higher distances than samples synthesized by a diffusion model.
> We tested this on the Pong environment. Non-attack diffusion samples yield low DIRE distances (0.05 ± 0.15), while true states show higher distances (0.61 ± 0.77), consistent with DIRE's assumptions. However, SHIFT-O, which introduces policy guidance during sampling, shifts the distribution further, achieving an even higher distance (0.94 ± 0.60). Although the higher mean might seem distinguishable, the high variance prevents reliable single-frame detection.
>
> As an example, here are five consecutive samples of DIRE distances (true state, SHIFT-O) from our experiment: [(0.59,0.63), (0.77,0.51), (0.72,1.23), (0.60,0.44), (0.60,0.48)]. Despite SHIFT-O having a higher average, 39% of true states throughout an episode with 2000 steps have a higher DIRE distance, making instance-level detection unreliable. Although it is still possible to use the mean DIRE distance to detect the existence of attacks, this requires aggregating statistics over a trajectory. By the time the attack is detected, it may have already influenced the victim’s decision-making and degraded its performance.
>
> These findings suggest that DIRE is not effective for detecting SHIFT attacks. More broadly, detection methods trained solely on clean data may struggle to identify adversarial samples since adversarial attacks may distort the distribution in an unexpected way.
>
> Q3a: Why was SA-MDP’s considered perturbation set not adapted to use, e.g., the reconstruction error from the auto-encoder, or some other notion better aligned with SHIFT?
>
> A3a: SA-MDP's $l_p$ norm constraint enforces spatial proximity but cannot capture SHIFT's multi-dimensional stealth requirements: realism, history alignment, and trajectory faithfulness. These properties are inherently interdependent yet distinct—no single metric (like reconstruction error) suffices. While reconstruction error partially addresses realism, it ignores history alignment and trajectory faithfulness. Forcing SHIFT into SA-MDP’s framework would sacrifice its core innovations for theoretical convenience.
>
> Q3b: Finetuned DMBP against SHIFT?
>
> A3b: As long as the attacker has the knowledge of how the victim finetune the diffusion model, the attacker can mimic the same process to generate the diffusion model as its new conditional diffusion base and reapply policy/realism guidance.
>
> Q4: Questions for the trilemma.
>
> A4: The trilemma is an empirical observation, not coming from SHIFT’s framework. We do not rule out the possibility of future methods overcoming this trade-off. Existing attacks like PGD, MinBest, PA-AD, and attacks based on high-sensitivity directions (e.g., blur, rotation) prioritize historical alignment and trajectory faithfulness but lack semantic change. SHIFT-O achieves semantic change and trajectory faithfulness, with slightly lower historical alignment, while SHIFT-I achieves semantic change and historical alignment, with slightly lower trajectory faithfulness.
>
> Q5: Table 2 only covers the Freeway environment and DP-DQN defense.
>
> A5: We provide extra results for Pong and SA-DQN below. Since SA-MDP is a relatively weak defense, other attack baselines also perform well. In terms of the stealthiness metric, it shows the same trend as Table 2. Table 2 focuses on DP-DQN because it represents the strongest existing defense, robust against large-budget attacks like PGD-15/255.
>
> | Attack| Reward↓| Dev (%)↑| Recons.↓| Wass. (×10⁻³)↓ | SSIM↑| LPIPS↓|
> |----------------|-----------------|---------------|---------------|----------------|----------------|---------------|
> | PGD-1/255      | 21.0 ± 0.0      | 0 ± 0.0       | 1.86 ± 0.3    | 1.33 ± 0.4     | 0.9737 ± 0.012 | 0.0876 ± 0.0403 |
> | PGD-15/255     | -21.0 ± 0.0     | 73.6 ± 8.6    | 3.43 ± 0.2    | 5.03 ± 0.4     | 0.7408 ± 0.026 | 0.3952 ± 0.0436 |
> | Transform(1,0) | -21.0 ± 0.0     | 62.4 ± 1.7    | 5.53 ± 0.2    | 1.18 ± 0.1     | 0.9045 ± 0.0074 | 0.0759 ± 0.0061 |
> | SHIFT-O-1.0    | **-21.0 ± 0.0** | **64.8 ± 0.7** | **0.77 ± 0.3** | **1.24 ± 0.2** | **0.9884 ± 0.013** | **0.0647 ± 0.0924** |
> | SHIFT-I-1.0    | **-20.8 ± 0.4** | **45.7 ± 3.0** | **0.88 ± 0.3** | **1.21 ± 0.3** | **0.9799 ± 0.016** | **0.0805 ± 0.0777** |
>
> Q6: What architecture and training schema is used for autoencoders? Why only perform a single step and step size 1 in the realism guidance? Resources used by GRAD?
>
> A6: The autoencoder uses 3 convolutional layers followed by a linear layer; the decoder mirrors this structure. It is trained with the Adam optimizer and MSE loss. We use a single realism guidance step with step size 1 as a balanced choice—small enough to preserve sample quality, yet large enough to enhance realism. A single step also ensures efficiency, as multiple steps would significantly increase computing time during SHIFT generation. GRAD uses V100 GPU and 20 hours for training defenses for environments such as Hoppers and Walkers, which does not use images as states. Thus, it is hard to migrate GRAD to the image domain due to computational overhead. We only have a single RTX 3090 and the training time for SHIFT is around 3-4 hours.
>
> Q7: What is the main real-world use case for SHIFT? In what situations can an attacker fully control the victim’s visual input? How is it realistic or relevant in practice?
>
> A7: SHIFT applies to real-world control tasks with visual input, such as robotics and autonomous driving. It assumes control over the agent’s observations, which is a common setup in prior work (e.g., SA-MDP, PA-AD). We also evaluate SHIFT under limited budgets (Appendix D.6). This is not a true best-case for the attacker, as SHIFT only alters observations, not actions. A stronger attacker would have total control of the victim and could directly manipulate the agent’s actions, which we do not assume.
>
> Q8: Questions related to semantic changed states effectiveness.
>
> A8: Attack effectiveness means the perturbed states cause the agent to take lower-reward actions. For example, in Figure 2, SHIFT generates semantically altered states that mislead the car into driving through a crosswalk with a pedestrian, causing a collision and reducing reward. While non-semantic attacks like PGD can degrade DQN performance, they fail against stronger defenses like diffusion-based DP-DQN (Table 2), where SHIFT remains effective.
>
> A9: During the actual testing episode of the state perturbation attack, the victim agent only sees the perturbed states and has no access to the true states. Therefore, the victim cannot compute SSIM nor LPIPS.
>
> A10: There may be multiple valid projection points, so we introduce D(⋅) to provide a mathematically rigorous and well-defined definition.
>
> A11: The reference policy is used to collect clean trajectories for training the diffusion model with some random actions (i.e., exploration) to ensure the data covers a diverse range of trajectories, including both low and high reward cases. Also see A1 to Reviewer u3xt.
>
> A12: Black et al. use reinforcement learning, an optimization method, to train a diffusion model, which is a generative model. This bridges generative modeling and optimization-based training.
>
> We hope our detailed rebuttal address could your concerns.

---

> > ### Comment · Reviewer_3Yfq · 2025-08-04
> > **Response to Rebuttal**
> >
> > Thank you for the extensive response. Please find further comments below regarding each response:
> >
> > A1a: Thank you for the clarification, the additional information in A1b and A2 is helpful to distinguish SHIFT’s notion of stealthiness from others in the literature. My remaining concern is the allusion to stealthiness towards humans. As you mentioned in your response to reviewer vqkj, the focus is on real-time evasion, hence SHIFT does not need to be stealthy towards humans (and as pointed out by reviewer vqkj, it is also not stealthy towards humans in general). In short, it is misleading to point out that existing methods (e.g., [31]) are not stealthy towards humans, since SHIFT may not be stealthy towards humans either, and the goal is to be stealthy towards the victim agent itself (and defenses).
> >
> > A1b: Thank you for pointing out the connection between Figure 6a and Table 2, while it indeed alleviates the issues with evaluating realism through the reconstruction error, I think this connection must be explicitly addressed in the text. Currently there is no information as to what environment was used for Figure 6a, how the 1.06+/- 0.5 reconstruction error is derived from the plot, and crucially, how it should be compared to the baseline results in Table 2.
> >
> >  A2: Thank you for sharing these insightful results into the detection of SHIFT perturbations using DIRE. Indeed, single instance detection with DIRE seems ineffective. I think these results and the discussion on the tradeoff of using DIRE on a trajectory or over multiple instances (which might work better but present some drawbacks) strengthens the claims that SHIFT is effective even in the presence of defenses. It would be also interesting to discuss potential benefits and drawbacks of incorporating DIRE or other defenses when computing SHIFT perturbations to make them adaptive.
> >
> > Could you please address the question regarding the proposed MAD and CUSUM detectors? What is the justification for choosing them and their specific parameters?
> >
> > Finally, could you please specify how the non-attack diffusion samples were generated for the experiments with DIRE?
> >
> >  A3a/b: Thank you for your response, the concerns regarding Q3 have been largely addressed by A2 in the rebuttal. However, I would argue that using the reconstruction error instead of an Lp norm for SA-MDP/DQN  would provide a fairer assessment of the defense’s performance against SHIFT, without modifying SHIFT itself to fit into the defense’s ideal setting. Therefore, it would be important to discuss how the use of Lp-norm based defenses might impact the evaluation (see also A5 below).
> >
> > A4: Thank you for the clarification. It would be important to rephrase this in the text, as line 332 currently reads “Our framework reveals a fundamental trilemma...”. It would also be good for clarity to specify that the “two properties” optimized by other methods are historical alignment and trajectory faithfulness in lines 333-334 and 408-410. Most importantly, could you please provide a more concrete answer regarding Q4b? I wonder if SHIFT or the baseline methods would have to e.g., reduce trajectory faithfulness to improve historical alignment.
> >
> > A5: Thank you for sharing these additional results for the Pong environment and SA-DQN defense, this alleviates my concerns on basing the main evaluations exclusively on the Freeway environment with DP-DQN. I think including your justification for using DP-DQN in the paper would significantly strengthen the results. My remaining concern here is that just like DP-DQN, SA-DQN is also designed to defend against norm-bounded attacks, therefore it would help to highlight how this influences the evaluation. I think the use of perturbations from [31], and further comparisons using them in the appendix, as well as the results introduced in A2 are important to highlight the fairness of the proposed evaluations.
> >
> > A6 and A11: Thank you for providing these implementation details, it would be important to include them in the text to provide a full picture of SHIFT’s architecture, training process, and computational complexity.
> >
> > A7: While indeed an attacker that controls the agent’s actions would be even more powerful, SHIFT can perform unconstrained perturbations, unlike most other attacks that also focus on perturbing the victim’s observations, hence it would be important to clarify that line 104 refers to a “worst-case scenario” from the perspective of the victim, as this could be misinterpreted because the paper proposes an attack, not a defense. Other than that, thanks to the reviewers for the responses regarding the practical relevance of SHIFT’s threat model in the rebuttal, which have been helpful.
> >
> > A8, A9, and A10: Thank you for clarifying these sentences, regarding A8 and A9, I would suggest a more concrete phrasing in the text.  In particular, I think it is important to avoid the suggestion that semantic change is needed for attack effectiveness (A8).

---

> ### Author Response · Authors · 2025-08-05
> **Further Clarifications**
>
> Dear Reviewer 3Yfq,
>
> Thanks for carefully reading our rebuttals and we will address your remaining concerns one by one.
>
> Q1a: Concerns about being stealthy toward humans.
>
> A1a: When only a single frame is available, both SHIFT-O and SHIFT-I appear realistic and plausible to human observers, due to the visual fidelity provided by diffusion models. However, high-sensitivity attacks (e.g., rotation) reported in [31] are easily noticeable to humans even in single-frame settings. When multiple frames are available, however, SHIFT-O becomes less stealthy to humans with domain knowledge because of its relatively weak historical alignment, whereas SHIFT-I maintains stealthiness by generating consistent illusionary trajectories. If true states can be queried, however, SHIFT-I can also be detected, as its generated trajectories may diverge significantly from the true state.
>
> Q1b: Setting of Figure 6a and how to compare with Table 2.
>
> A1b: Figure 6a was conducted in the Freeway environment using a vanilla DQN policy under SHIFT-O attack, and the 1.06(0.5) reconstruction error was derived from the raw data. Hence, it is not directly comparable to Table 2, which used DP-DQN as defenses. Following the reviewer’s suggestion, we reran the experiments in the Freeway environment under the DP-DQN policy without realism guidance and report the results: SHIFT-O: 1.07 (0.4), SHIFT-I: 1.11 (0.5). These results are now under the same setting as Table 2 and are directly comparable to all other baselines.
>
> Q2: Question regarding DIRE and MAD and CUSUM detectors.
>
> A2: Our SHIFT attacks can adapt to various defense methods by using the defense policies to guide the generation of perturbed states. However, DIRE is different from typical defenses that train a robust policy mapping from states to actions. Instead, DIRE is a detection method for states that are generated by a diffusion model. Thus, SHIFT cannot directly use DIRE for policy guidance. But, it is possible to incorporate DIRE distance into SHIFT, similar to our realism enhancer, by minimizing DIRE distance. However, this requires additional designs and experimentation, making it an interesting future work. For the DIRE experiment, we used our diffusion model conditioned on true historical states and actions to generate clean diffusion samples.
>
> Both MAD and CUSUM operate on Wasserstein-1 distances between consecutive observed states and use statistics from clean trajectories as baseline detection metrics. For MAD, the baseline is computed as median + 5 (threshold) × MAD, which is 0.00116 in the Freeway environment. MAD flags an anomaly when the Wasserstein distance exceeds this baseline for three consecutive steps. CUSUM employs a more complex calculation (details in [41]). With drift = 1.5 and threshold = 3, the resulting baseline in Freeway is 0.00185. CUSUM triggers detection when the cumulative sum of deviations surpasses this baseline. For both methods, we set the hyperparameters (threshold and drift) to the lowest values that ensure clean trajectories remain undetected.
>
> Q3/5: Concerns regarding the evaluations
>
> A3/5: We note that DP-DQN is not tailored to either $l_p$-norm-based or diffusion-based attacks. It uses a diffusion model to purify perturbed states, forms a belief about possible true states, and applies a pretrained max–min strategy to choose robust actions. Its robustness is not specific to any attack type. The DP-DQN paper only tested $l_p$-norm attacks because no well-established attacks beyond $l_p$-norm constraint existed at the time. Thus, using DP-DQN as a defense to evaluate SHIFT and other attacks is fair.
>
> Further, as discussed in A3 of our response to Reviewer vqkJ, designing a strong defense capable of mitigating SHIFT is non-trivial. Even combining reconstruction error with the SA-MDP framework would require substantial investigation, since reconstruction error from an autoencoder cannot be directly measured or clipped in the same way as an lp​-norm constraint. Developing such a defense remains a promising direction for future work.
>
> Q4: Trade-off between different attack properties
>
> A4: Our “trilemma” observation is mainly empirical and is derived from our evaluation of SHIFT and other attacks, but we do not have a formal proof that a trade-off must exist. In our experiments, a trade-off emerges for our semantics-changing SHIFT-O and SHIFT-I attacks. As shown in Table 2, SHIFT-O attains higher trajectory faithfulness, while SHIFT-I achieves stronger historical alignment. However, this pattern does not hold for PGD, which does not produce essential semantic changes under a small budget. Shrinking the PGD budget from 15/255 to 1/255 improves both historical alignment and trajectory faithfulness. These results suggest that when semantic changes are present, a trade-off between historical alignment and trajectory faithfulness would occur.
>
> We thank you again for your valuable feedback, and we will integrate all clarifications into our manuscript.

---

> > ### Comment · Reviewer_3Yfq · 2025-08-06
> > **Response to clarifications**
> >
> > A1 and A2: Thank you for the further clarifications and results, they have addressed my concerns regarding the corresponding sections in the paper/appendix.
> >
> > A4: Thank you for the more concrete description of the trilemma, the details provided during the rebuttal have alleviated my concerns with this part of the paper.
> >
> > A3/5: Thank you for the follow-up discussion, it is understandable that adjusting DP-DQN/SA-MDP to defend against SHIFT (or even the attacks by [31]) is not trivial,  and that DP-DQN does not necessarily use Lp norm-bounded attacks for its proposed pessimistic policy. However, I would still argue that if the instances of these attacks used for evaluation do consider Lp
> >  norm-bounded attacks, mentioning how this may or may not be advantageous for SHIFT (and for the attacks by [31], which are also not bounded in an Lp norm) would allow for a more complete interpretation of the numerical results.
> >
> > Thank you once again for the thorough rebuttal.

---

> > > ### Author Response · Authors · 2025-08-08
> > > **Have We Addressed All Your Concerns?**
> > >
> > > Dear Reviewer 3Yfq,
> > >
> > > As the discussion period draws to a close, we would like to kindly ask if our last responses have successfully addressed all of your concerns. If so, we would greatly appreciate it if you could consider updating your final rating to reflect our discussion.
> > > Thank you once again for your time and thoughtful feedback.
> > >
> > > Sincerely,
> > >
> > > Authors of Submission 9986

---

> ### Author Response · Authors · 2025-08-06
> **Further Clarifications (2)**
>
> Dear Reviewer 3Yfq:
>
> Thanks for the follow-up questions and we will respond to your last concern here.
>
> Q1: Relations between non-$l_p$-norm-based attacks (SHIFT and [31]) and $l_p$-norm-based defenses
>
> A1: We agree that it is important to highlight that current $l_p$-norm-based defenses, such as SA-DQN and WocaR-DQN, are specially designed to handle state perturbations constrained within a fixed $l_p$-norm distance. Our SHIFT attacks, as well as the attacks presented in [31], operate outside of this framework and exploit this limitation. These non-$l_p$  attacks can bypass $l_p$-norm-based defenses because the robustness regularizers used during their training do not account for perturbations beyond the $l_p$ constraint.
>
> In our evaluation, we also included two diffusion-based defenses, DMBP and DP-DQN, which are not limited by $l_p$-norm assumptions. These defenses aim to reconstruct clean observations and are therefore more robust to a broader range of perturbations. As shown in Table 2 and Appendix D.8, the attack from [31] is largely ineffective against these diffusion-based defenses, because it does not introduce meaningful semantic changes. In contrast, to the best of our knowledge, SHIFT is the only attack that successfully bypasses these diffusion-based defenses.
>
> Additionally, as noted in A1 of our previous response, the attacks from [31] are more perceptible to humans in single-frame settings, making them less stealthy than SHIFT.
>
> We hope this response fully addresses all your remaining concerns. If so, we would greatly appreciate it if you could consider updating your final rating to reflect our discussion.
>
> Sincerely,
>
> Authors of Submission 9986

---

### Official Review · Reviewer_2wp2 · 2025-06-29

**Clarity:** 3
**Significance:** 3
**Originality:** 3
**Rating:** 4
**Confidence:** 4

**Summary:**

The paper introduces a novel attack method called SHIFT, which leverages diffusion models to generate adversarial state perturbations in reinforcement learning (RL) systems, particularly in vision-based environments.  SHIFT is a policy-agnostic, diffusion-based attack framework designed to generate perturbed states that are semantically different from the true states while remaining realistic and aligned with the agent's history. The attack is grounded in three key properties: semantic-altering, historically-aligned, and trajectory-faithful. SHIFT uses classifier-free guidance to generate history-aligned states and policy guidance to induce semantic changes, ensuring the perturbations are both effective and stealthy.

**Questions:**

See weaknesses. Overall, the paper presents a significant advancement in the field of adversarial attacks on RL systems by introducing SHIFT, a powerful and stealthy diffusion-based attack method. The strengths of the paper lie in its innovative approach, comprehensive evaluation, and theoretical contributions. However, the computational complexity, limited scalability of SHIFT are areas that need further improvement.

**Ethical Concerns:**

["NO or VERY MINOR ethics concerns only"]

**Final Justification:**

Thanks for the authors' responses.

According to the rebuttal, training diffusion models does consume excessive resources and has certain scalability issues.

However, I  believe this does not affect the technical insights and contributions of this paper. With the rapid development of diffusion models, I remain optimistic that this limitation will be resolved.

In light of the above points, I maintain my original score and am inclined to give this paper a positive evaluation.

**Quality:**

3

**Strengths And Weaknesses:**

Strengths:

1. The paper introduces SHIFT, a novel diffusion-based adversarial attack method that goes beyond traditional $l_p$-norm constraints. This approach leverages the power of diffusion models to generate semantically meaningful perturbations, which is a significant advancement in the field of adversarial attacks on RL systems.

2. SHIFT focuses on generating perturbations that alter the semantics of the input states, which is a critical aspect often overlooked by existing attacks. By changing the meaning of the input, SHIFT can significantly impact the agent's decision-making process.

3. The authors conduct extensive experiments across multiple environments, including Atari games, Doom, and an AirSim autonomous driving simulator. This demonstrates the generalizability of SHIFT across different types of RL tasks and environments. SHIFT is tested against several state-of-the-art defenses, including regularization-based methods and diffusion-based defenses. The results show that SHIFT can effectively break these defenses, highlighting its potency.


Weaknesses:

1. Computational Complexity. The use of diffusion models and the need for classifier-free and policy guidance introduce significant computational overhead. Training the conditional diffusion model and the autoencoder-based anomaly detector can be resource-intensive, which may limit the practical applicability of SHIFT in real-time scenarios.

2. Sampling Efficiency. While the authors mention using the EDM formulation to improve sampling efficiency, the overall computational cost of generating perturbations in real-time remains a concern. This could be a barrier for deploying SHIFT in environments that require rapid decision-making.

3. Limited Scalability. The diffusion model needs to be trained separately for each distinct environment, which can be impractical for large-scale deployment. This limits the scalability of SHIFT, especially in scenarios where the RL agent operates in multiple or dynamically changing environments.

---

> ### Author Rebuttal · Authors · 2025-07-31
>
> Dear Reviewer 2wp2:
>
> Thanks for your insightful feedback, and we will address your concerns in the following.
>
> Q1: Computational Complexity. The use of diffusion models and the need for classifier-free and policy guidance introduce significant computational overhead. Training the conditional diffusion model and the autoencoder-based anomaly detector can be resource-intensive, which may limit the practical applicability of SHIFT in real-time scenarios.
>
> A1: Although SHIFT may incur a certain level of computational overhead, training the diffusion model takes around 4 hours and the autoencoder-based anomaly detector takes about 1 hour on the Freeway environment—both of which can be done in parallel. In comparison, other learning-based attack methods such as PA-AD[1] require around 12 hours on the same hardware to train an attack policy on Freeway. Therefore, SHIFT requires significantly less training time than PA-AD, a state-of-the-art learning-based attack method.
>
> On the defense side, prior defense methods such as SA-DQN and WocaR-DQN require more than 24 hours to train the defense policies, which is longer than the training time of our SHIFT attack. This creates a sufficient time window for SHIFT to inject adversarial samples before such defenses are deployed.
>
> [1] Sun et al., Who Is the Strongest Enemy? Towards Optimal and Efficient Evasion Attacks in Deep RL. ICLR 2022
>
> Q2: Sampling Efficiency. While the authors mention using the EDM formulation to improve sampling efficiency, the overall computational cost of generating perturbations in real-time remains a concern. This could be a barrier for deploying SHIFT in environments that require rapid decision-making.
>
> A2: While SHIFT may introduce a moderate sampling overhead, we emphasize that all experiments were conducted using a single RTX 3090 GPU. Despite this relatively limited computing resource, SHIFT achieves practical sampling times (approximately 0.2 seconds per sample). With more powerful hardware, the efficiency of SHIFT could be further improved, making real-time deployment increasingly feasible.
>
> Q3: Limited Scalability. The diffusion model needs to be trained separately for each distinct environment, which can be impractical for large-scale deployment. This limits the scalability of SHIFT, especially in scenarios where the RL agent operates in multiple or dynamically changing environments.
>
> A3: Thank you for highlighting this important scalability concern. In our current setup, we do train a separate conditional diffusion model for each environment. However, a promising direction for future work is to train a single conditional diffusion model across multiple environments by leveraging the history conditioning mechanism. The history input provides contextual cues that help the model infer which environment it is operating in, preventing mode collapse or sample confusion.
>
> A similar idea has been explored in TD-MPC2 [2], which learns transition dynamics across multiple environments in a shared latent space. Their results show that a single model can handle multiple tasks simultaneously. We could extend this approach by adopting latent diffusion models and performing attack guidance in the latent space, allowing a single diffusion model to support various environments. This strategy would significantly improve SHIFT’s scalability, making it more suitable for complex or dynamically changing environments.
>
> [2] Hansen et al., TD-MPC2: Self-Supervised Decision Making with Latent World Models, NeurIPS(2023)
>
> We hope our detailed rebuttal could address most of your concerns. If you have further questions please let us know and we will try our best to clarify them.
>
> Sincerely,
>
> Authors of Submission 9986

---

> > ### Comment · Reviewer_2wp2 · 2025-08-01
> > **Rebuttal Acknowledgments**
> >
> > Thanks for the authors' responses.
> >
> > According to the rebuttal, training diffusion models does consume excessive resources and has certain scalability issues.
> >
> > However, I believe this does not affect the technical insights and contributions of this paper. With the rapid development of diffusion models, I remain optimistic that this limitation will be resolved.
> >
> > In light of the above points, I maintain my original score and am inclined to give this paper a positive evaluation.

---

### Official Review · Reviewer_u3xt · 2025-06-30

**Clarity:** 3
**Significance:** 3
**Originality:** 3
**Rating:** 4
**Confidence:** 2

**Summary:**

This paper presents novel characterizations of semantics-aware and stealthy adversarial attacks in sequential decision-making settings, introducing three key principles: Semantic Change, Historical Alignment, and Trajectory Faithfulness. Building on these insights, the authors propose a novel, policy-agnostic attack framework based on diffusion models, which perturbs visual observations beyond traditional ℓₚ-norm-constrained methods. The proposed approach generates semantically altered adversarial perturbations that are less detectable. Empirical results show that the method outperforms existing attack baselines.

**Questions:**

1. In lines 242–243, is an offline dataset collected from a clean environment required, or is it necessary to generate trajectory data by interacting with the clean environment using $\pi_{ref}$ to assist training? What are the requirements for $\pi_{ref}$? Does it need to ensure a certain level of coverage over the state or trajectory distribution?

2. Why does SHIFT-I not incorporate information about $a_{t-1}$? What would the performance of SHIFT-I be if $a_{t-1}$ were included? Similarly, how would the performance of SHIFT-O be affected if it also omitted $a_{t-1}$?

3. Regarding the experiments in Table 2, how would the performance of SHIFT be affected if the attack frequency were reduced, for example, in variants like SHIFT-I-0.25 and SHIFT-O-0.25?

4. The generation time of SHIFT is approximately 0.2 seconds. It remains unclear whether, in practice, the agent can detect the presence of an attack by observing differences in generation times of subsequent states between normal (no-attack) and SHIFT attack scenarios. Furthermore, it is of interest to investigate how increasing the generation iterations of methods such as PGD, PA-AD-TC, MinBest, and PA-AD, or decreasing the number of reverse steps in SHIFT—thereby aligning their adversarial state generation times to the same scale—would affect their comparative performance.

5. There is a typo in line 257; it should be Figure 6c.

**Ethical Concerns:**

["NO or VERY MINOR ethics concerns only"]

**Final Justification:**

The authors clearly explained the data collection and experimental setup, supplemented the explanation and experimental verification of the difference between SHIFT-I and SHIFT-O, and the experimental results of SHIFT variants. The authors addressed my inquiry regarding generation time. As all my major concerns have been properly resolved, I maintain my original score.

**Limitations:**

Please refer to the Weaknesses and Questions.

**Paper Formatting Concerns:**

There is no format problem with this paper.

**Quality:**

3

**Strengths And Weaknesses:**

The paper is motivated by a reasonable intuition, proposes a novel method, and provides empirical results that suggest its effectiveness.

Strengths：

1.	The paper is well-organized, and the concepts are clearly articulated.

2.	The proposed three properties are intuitive and well-motivated.

3.	The proposed method is novel.

4.	Experimental results demonstrate the effectiveness of the proposed approach.

Weaknesses：

1.	The time complexity of SHIFT is relatively high, with a generation time of approximately 0.2 seconds, about 10 times longer than that of methods such as PGD, which typically require around 0.02 seconds.

---

> ### Author Rebuttal · Authors · 2025-07-31
>
> Dear Reviewer u3xt:
>
> Thanks for your insightful feedback and we will address your concerns in the following.
>
> Q1: In lines 242–243, is an offline dataset collected from a clean environment required, or is it necessary to generate trajectory data by interacting with the clean environment using $\pi_{ref}$ to assist training? What are the requirements for $\pi_{ref}$? Does it need to ensure a certain level of coverage over the state or trajectory distribution?
>
> A1: Yes, SHIFT requires collecting sufficient trajectory data from a clean environment to train the conditional diffusion model. To this end, we use a pre-trained reference policy $\pi_{ref}$ that performs well in the unperturbed environment. To ensure adequate coverage over the trajectory distribution, we incorporate a small amount of random actions (i.e., exploration) during data collection. This helps the conditional diffusion model learn not only high-reward behavior but also the broader dynamics of the environment, including suboptimal trajectories.
>
> Q2: Why does SHIFT-I not incorporate information about $a_{t-1}$? What would the performance of SHIFT-I be if $a_{t-1}$ were included? Similarly, how would the performance of SHIFT-O be affected if it also omitted ${a_{t-1}}$?
>
> A2: This is a great question. In SHIFT-I, we deliberately omit $a_{t-1}$ to relax the conditional generation space, allowing the diffusion model to search for perturbed states that better align with the observed history. Including $a_{t-1}$ in SHIFT-I would overly constrain the generation distribution, introducing subtle inconsistencies—similar to those observed in SHIFT-O—that harm historical alignment.
> Empirically, including $a_{t-1}$ in SHIFT-I results in an attack performance of 11.3 (4.1) in the Freeway environment against DP-DQN, which is comparable to the original result of 10.6 (2.5). However, stealthiness is reduced: the Wasserstein distance of SHIFT-I increases from $0.84 (0.2) × 10^{-3}$ to $0.90 (0.2) × 10^{-3}$, indicating degraded historical alignment. Combined with SHIFT-I’s inherently lower trajectory faithfulness, this makes the attack significantly less stealthy.
>
> On the other hand, omitting $a_{t-1}$ from SHIFT-O improves attack performance (from 21.8 (3.2) to 14.1 (2.8)) in the Freeway environment against DP-DQN but at the cost of trajectory faithfulness: SSIM drops from 0.9990 (0.0014) to 0.9915 (0.0015), and LPIPS increases from 0.0008 (0.0014) to 0.0121 (0.001).
>
> In summary, the inclusion or omission of $a_{t-1}$ reflects a trade-off between historical alignment and trajectory faithfulness. Each SHIFT variant is designed to balance this trade-off in accordance with its design goal.
>
> Q3: Regarding the experiments in Table 2, how would the performance of SHIFT be affected if the attack frequency were reduced, for example, in variants like SHIFT-I-0.25 and SHIFT-O-0.25?
>
> A3: We provide an ablation study on different attack frequencies—including 1.0, 0.5, 0.25, and 0.15—for both SHIFT-I and SHIFT-O in Appendix D.6. The results show that even at lower frequencies such as 0.25, SHIFT maintains a strong attack performance, demonstrating its effectiveness under limited perturbation budgets.
>
> Q4: The generation time of SHIFT is approximately 0.2 seconds. It remains unclear whether, in practice, the agent can detect the presence of an attack by observing differences in generation times of subsequent states between normal (no-attack) and SHIFT attack scenarios. Furthermore, it is of interest to investigate how increasing the generation iterations of methods such as PGD, PA-AD-TC, MinBest, and PA-AD, or decreasing the number of reverse steps in SHIFT—thereby aligning their adversarial state generation times to the same scale—would affect their comparative performance.
>
> A4: The 0.02 seconds reported for PGD, MinBest, and PA-AD are based on 10 steps of gradient descent. As shown in the SA-MDP paper[1], using 50 steps is also common. We tested these methods with 50 steps in Freeway under DP-DQN defense. In terms of attack performance,  PGD(1/255) with 50 steps has 29.8(1.0) compared with 10 steps 30.0(0.9) and PGD(15/255) with 50 steps has 29.2(0.8) compared with 10 steps 29.0(1.0). This is mainly because even with 10 steps, perturbation has reached the budget limit under PGD attack. While the generation time increased to an average of 0.1 seconds, their attack performance did not improve, as they still failed to bypass DP-DQN. In this setting, SHIFT takes about twice the time but offers significantly stronger attack performance. However, it is challenging to reduce SHIFT's generation time further—reducing the number of reverse steps in SHIFT would degrade the quality of adversarial samples. That said, our experiments were conducted under limited compute resources (a single RTX 3090). With more powerful hardware, the sampling time of SHIFT could potentially be further reduced.
>
> We hope our detailed rebuttal could address most of your concerns. If you have further questions please let us know and we will try our best to clarify them.
>
> Sincerely,
>
> Authors of Submission 9986
>
> [1] Zhang et. al., Robust Deep Reinforcement Learning against Adversarial Perturbations on State Observations. NeurIPS 2020.

---

> > ### Comment · Reviewer_u3xt · 2025-08-01
> >
> > Thank you very much for your response. It clarified my concerns, and I will maintain my positive rating.

---

### Official Review · Reviewer_vqkJ · 2025-07-02

**Clarity:** 3
**Significance:** 3
**Originality:** 3
**Rating:** 5
**Confidence:** 4

**Summary:**

This paper aims to generate adversarial attacks for RL systems via state perturbations, while restricting the perturbation to remain in the distribution of realistic trajectories given the agent's history. It does this through training a history-conditioned diffusion model to generate plausible next states and using classifier guidance to search for next states which would cause the policy to perform worse. In addition, it restricts this search with an auto-encoder-based method of anomaly detection to maintain a greater degree of realism.

**Questions:**

How could we expect a system to be robust to these sorts of attacks?

In what scenario would an attacker have the ability to execute this attack while not having the means and inclination to do a much more direct attack? (like taking over the steering directly)

**Ethical Concerns:**

["NO or VERY MINOR ethics concerns only"]

**Final Justification:**

The approach used in this paper is interesting and useful, as it shows how to generate semantically meaningful adversarial attacks on a policy that will be useful in many domains that need high-dimensional adversarial search. However, I believe that the particular attack model they are working with is not likely, since it proposes a very specific sort of access from an attacker (full control over sensors and no other control). Moreover, even if the attacker had this specific sort of access it seems impossible to defend against, so it is hard to understand the utility (for the defence) of generating better attacks.

**Limitations:**

I believe the technical limitations are adequately addressed.

**Paper Formatting Concerns:**

no major concerns

**Quality:**

3

**Strengths And Weaknesses:**

Strengths:
* The approach of training a diffusion model and using classifier guidance to produce adversarial attacks is interesting and broadly effective
* The regularisation with auto-encoder-based anomaly detection is an interesting and powerful trick to get this method to work.
* It appears to work as intended. Figure 2 is a clear example of the method working.
* Adversarially searching through synthetic but semantically valid generations is a broadly useful problem and it was previously unclear to me how this could be achieved. I would expect this to be useful in adversarial evaluation broadly and in unsupervised environment design.

Weaknesses:
* The setting is unrealistic as an adversarial attack. If the attacker had this much control over the observations they would likely have access to control the hardware directly and would not need to resort to this level of complexity.
* In addition, it is not clear how making it look semantically reasonable makes the attack undetectable. It is quite noticeable that the pedestrian has been removed from the image in Figure 2. If both videos are saved for audit, or if a human were watching, it would be quite visible.
* Further, if we were aiming to use this for adversarial training it is unclear what we would want the agent to do. Ultimately if the image looks indistinguishable from a clear road ahead I can see no way that any system would be able to avoid hitting pedestrians. This is why perturbations for ML models in adversarial attacks have been norm-ball restricted, because we aim to preserve the semantics of the image as to find situations where the agents decision changes when it should not. In the situations generated in this paper, the behavior of the system ought to change.
* The theoretical formalism does not quite match what is being done.
	* The definition of "valid states" would preclude the sorts of semantically sensible generations that we see from diffusion models at scale which are combinatorial generalisations of realistic images like "an astronaut riding a horse on the moon". Given that this technique is based on diffusion models it would likely allow for these sorts of combinatorial generalisations but they are disallowed by definition 3.1.
		* This critique is inherited by the definition of "realistic states"
	* More concerning is that the definition of semantics-changing states does not quite match the name. There are semantically identical images that are quite far in pixel space, only changing the texture, color, hue, or other irrelevant aspects of the image. I would have expected "semantics-changing" to involved aspects of the image which would change the policy.
* The approach should cite works from the generative environment literature, as it is essentially training a generative world/environment model. In essense, this can be seen as an application of a generative world model, or a way of leveraging a pre-trained generative world model for adversarial search.

---

> ### Author Rebuttal · Authors · 2025-07-31
>
> Dear Reviewer vqkJ,
>
> Thank you for your insightful feedback. We will address your concerns in the following.
>
> Q1: The setting is unrealistic as an adversarial attack. If the attacker had this much control over the observations, they would likely have access to control the hardware directly and would not need to resort to this level of complexity. In what scenario would an attacker have the ability to execute this attack while not having the means and inclination to do a much more direct attack? (like taking over the steering directly)
>
> A1: The state perturbation attack setting assumes that the attacker can manipulate only the observations received by the victim agent, thereby misleading it into choosing suboptimal actions. This assumption is standard in prior literature, including SA-MDP[1] and PA-AD[2]. We argue that gaining access to an agent’s sensory input (e.g., camera or radar) is considerably more feasible than gaining full control over its actuation system. For example, in autonomous driving, it may be easier for an attacker to hack into perception components, such as cameras or LiDAR, to subtly manipulate visual inputs, rather than breach the entire control system to directly steer the vehicle.
>
> Moreover, our attack is relevant to real-world robotic and autonomous systems that rely on visual observations. SHIFT can be deployed in such scenarios to induce subtle, semantically meaningful perturbations without needing full system control.
>
> [1] Zhang et al., Robust Deep Reinforcement Learning against Adversarial Perturbations on State Observations. NeurIPS 2020.
>
> [2] Sun et al., Who Is the Strongest Enemy? Towards Optimal and Efficient Evasion Attacks in Deep RL. ICLR 2022.
>
> Q2: In addition, it is not clear how making it look semantically reasonable makes the attack undetectable. It is quite noticeable that the pedestrian has been removed from the image in Figure 2. If both videos are saved for audit, or if a human were watching, it would be quite visible.
>
> A2: As mentioned in Q1/A1, during testing the victim agent only observes the perturbed states and has no access to the true states. Therefore, it cannot detect perturbed states or compare perturbed and true states in real time. Moreover, even if both true and perturbed trajectories are saved for future auditing, the attack would have already influenced the victim’s decision-making and degraded its performance before being detected. Our focus is on real-time evasion during deployment, where SHIFT remains hard to detect under typical assumptions in the state perturbation threat model.
>
> Q3: Further, if we were aiming to use this for adversarial training it is unclear what we would want the agent to do. Ultimately if the image looks indistinguishable from a clear road ahead I can see no way that any system would be able to avoid hitting pedestrians. This is why perturbations for ML models in adversarial attacks have been norm-ball restricted, because we aim to preserve the semantics of the image as to find situations where the agents decision changes when it should not. In the situations generated in this paper, the behavior of the system ought to change.
>
> A3: This is an insightful point, and we agree that applying SHIFT for adversarial training is a promising future direction. Compared to traditional $l_p$ norm-based perturbations, SHIFT induces semantic changes, allowing us to go beyond simple pixel-level perturbations.
> Instead of assuming small norm perturbations that preserve semantics, agents under SHIFT can be trained to form a semantic belief about the underlying true state and make robust decisions based on this belief when encountering perturbed observations.
> We also acknowledge that in some scenarios, the semantic change may be too strong for the agent to act optimally,e.g., when a pedestrian is fully removed. However, this highlights the value of budget-aware sensing: if the agent has a limited probing budget (e.g., can occasionally access true observations), SHIFT can help identify high-risk situations in which the agent should query the environment more carefully.
> Thus, SHIFT can facilitate learning robust policies by revealing when perception failure is most consequential.
>
> Q4a: The definition of "valid states" would preclude the sorts of semantically sensible generations that we see from diffusion models at scale which are combinatorial generalisations of realistic images like "an astronaut riding a horse on the moon". Given that this technique is based on diffusion models it would likely allow for these sorts of combinatorial generalisations but they are disallowed by definition 3.1.
>
> A4a: The conditional diffusion model used by SHIFT is trained specifically on data collected from the target environment. For example, in the autonomous driving scenario, we train the model from scratch using trajectories recorded in clean, unperturbed driving environments.
> As a result, the trained diffusion model captures the semantics and physical constraints of the environment and does not generate out-of-distribution samples such as "an astronaut riding a horse on the moon."
>
> Q4b: More concerning is that the definition of semantics-changing states does not quite match the name. There are semantically identical images that are quite far in pixel space, only changing the texture, color, hue, or other irrelevant aspects of the image. I would have expected "semantics-changing" to involved aspects of the image which would change the policy.
>
> A4b: This is a great point. We agree that certain image-level changes, such as texture, brightness, or contrast, can appear large in pixel space but remain semantically identical. However, state-of-the-art diffusion-based defenses like DMBP and DP-DQN are specifically designed to denoise and project the input back onto the valid states. As shown in Appendix D.8, attacks based on high-sensitivity directions (e.g., blur, brightness changes) fail to affect these defenses.
>
> In contrast, SHIFT generates semantically altered states that can bypass diffusion-based denoisers and actually lead the agent to take different actions. Still, as shown in Appendix D.8, attacks based on high-sensitivity directions can change the policy behavior under weaker defenses or no defense at all.
>
> This leads to an important nuance: if we define “semantics-changing” solely by whether the agent’s action changes, the definition becomes policy-dependent. For example, brightness changes might alter the actions of a vanilla DQN agent but have no effect on a DP-DQN agent. To avoid such ambiguity, our definition of semantics-changing states (Def. 3.3) does not rely solely on policy behavior and does not exclude non-essential semantic changes—reflecting the broader goal of evaluating stealthy, history-aligned, and behavior-impacting perturbations.
>
> Q5: The approach should cite works from the generative environment literature, as it is essentially training a generative world/environment model. In essense, this can be seen as an application of a generative world model, or a way of leveraging a pre-trained generative world model for adversarial search.
>
> A5: Thanks for pointing out this line of related work and we will add a section in the related work in the revised version of our paper. We will include papers such as [3][4][5].
>
> [3] Hafner et al., Dream to Control: Learning Behaviors by Latent Imagination. ICLR 2020.
>
> [4] Pérez et al., EnvGen: Generating Diverse and Challenging Environment Variants. ICLR 2023.
>
> [5] Zhou et al., DINO World Model (DINO‑WM): World Models on Pre‑trained Visual Features Enable Zero‑Shot Planning. ICML 2025.
>
> We hope our detailed rebuttal could address most of your concerns. If you have further questions, please let us know and we will try our best to clarify them.
>
> Sincerely,
>
> Authors of Submission 9986

---

> > ### Comment · Reviewer_vqkJ · 2025-08-07
> > **Response to Rebuttal**
> >
> > I thank the authors for their detailed response. While I acknowledge that prior work has made similar assumptions of the attack I still disagree about it's feasibility in practice. Moreover, if the sensors were being completely controlled such that arbitrary inputs could be fed in, I struggle to understand the use of constructing these attacks when the defense is impossible.
> >
> > I believe there was a misunderstanding of the point of question Q4a. The main point is that the generations shown by the method are not close in pixel space to a true generation necessarily, just a plausible generation. This doesn't undermine the approach, it only means that the definition of "valid" here is overly narrow as to preclude generations of novel scenes that are physically possible and realistic but not previously seen in the training data. In addition, this sort of combinatorial generalization would happen within the domain as well, as it is just a feature of generative modeling.

---

> ### Author Response · Authors · 2025-08-08
> **Further Clarifications**
>
> Dear Reviewer vqkJ:
>
> Thanks for reading our rebuttal and we will clarify your further concerns.
>
> Q1: Totally controlling the sensors is not realistic in practice. Is there any chance to defend SHIFT?
>
> A1: While total control of the victim’s sensors is indeed a strong and often unrealistic assumption, we have considered more practical scenarios in our study. Specifically, we investigate the case where SHIFT operates under a limited budget, restricting the attacker to inject perturbations only during a fraction of the time steps. As shown in Appendix D.6, even with such constraints, SHIFT can still significantly degrade agent performance.
> Additionally, we acknowledge that attackers may only be able to perturb a subset of pixels in each state, which is more realistic than full sensor control. Our preliminary idea involves applying SHIFT within a masked region of the state, constrained by a pixel budget. Identifying an optimal mask under these constraints is non-trivial and represents a promising direction for future research.
>
> Regarding defenses, we do not rule out the possibility of defending against SHIFT. As mentioned in A3, agents could leverage historical information to build semantic beliefs about the true state, enabling robust decision-making even under perturbed observations. Moreover, the inherent signatures of diffusion models offer another avenue for defense. For instance, detectors like DIRE can identify images generated by diffusion models, even if they appear realistic to humans. While current methods are not yet effective against SHIFT (see our response to Reviewer 3Yfq), we believe that further exploration of diffusion model’s inherent signatures could lead to more efficient defenses.
>
> Q2: “Valid States” definition is too narrow.
>
> A2:  We appreciate the reviewer’s clarification regarding the concern of “valid states.” We agree that diffusion models can generate combinatorial generalizations not present in the training set. However, whether these generations are considered valid states depends on the underlying world model.
>
> To illustrate, consider the example “an astronaut riding a horse on the moon”:
> 1. If the world model is sufficiently general to simulate various humans riding different objects on different planets, the training data would include both humans riding horses on Earth and astronauts driving rockets on the moon. In this case, the diffusion model could generate “an astronaut riding a horse on the moon,” and since the world model can simulate this scenario, it would be considered a valid state.
>
> 2. If the world model is narrower and only simulates humans riding different objects on Earth, the training data would not include moon scenarios, and “an astronaut riding a horse on the moon” is very unlikely to be generated from a diffusion model traind from scratch using only data from this world model.
>
> 3. If the world model used in the first case incorporates real biological simulations—meaning, for example, that a horse cannot survive on the Moon. While the diffusion model might still generate an sample of “an astronaut riding a horse on the Moon”, such a scenario would be prohibited in the world model due to its adherence to biological constraints. As a result, “an astronaut riding a horse on the Moon” would be considered invalid state in this case.
>
> We also agree that there is no universally “correct” definition of valid states; different definitions may be appropriate for different tasks. In the context of SHIFT attacks, our goal is to maintain stealthiness to humans with domain knowledge of the relevant task. Defining valid states based on a narrower, task-specific world model enhances stealthiness, as humans are more likely to detect out-of-context scenarios (e.g., “an astronaut riding a horse on the moon” in a driving task) as attacks, while SHIFT-generated perturbations may remain undetected. On the other hand, broader definitions of valid states may be beneficial for tasks that require greater diversity, such as generative environments.
>
> We hope our further clarifications can address your concerns.
>
> Sincerely,
>
> Authors of Submission 9986

---

### Note · Authors · 2025-08-11

Dear AC and Reviewers,

Thank you once again for your time and effort in reviewing our paper and for providing insightful, constructive feedback. We would like to take this opportunity to summarize our work and formally conclude the author-reviewer discussion.

Our paper introduces SHIFT, a novel diffusion model-based attack framework that demonstrates strong attack performance against robust defenses while maintaining stealthiness. From the initial review, we received supportive feedback from Reviewers vqkJ, u3xt, and 2wp2, and we particularly value the detailed and constructive comments from Reviewer 3Yfq.

To summarize, all reviewers acknowledged the strengths of our work, including:

1. The clear motivation and novelty of the proposed SHIFT attack.

2. The extensive experiments conducted to demonstrate the effectiveness and stealthiness of SHIFT.

A common concern among reviewers pertained to the definitions of "stealthy" and "semantic changed states," which stem from the new semantics-aware attack framework we proposed.

During the rebuttal period, we provided detailed responses and additional experiments to address the concerns. We believe that we have thoroughly addressed all points raised by each reviewer. We especially appreciate the fruitful discussion with Reviewer vqkJ, which connected our SHIFT framework to broader topics in adversarial evaluation and unsupervised environment design. We are also deeply grateful to Reviewer 3Yfq for their ongoing constructive discussions, and we will incorporate the further clarifications and explanations from our rebuttal into the final revision of our paper.

Once again, we sincerely thank all reviewers and the AC for their thoughtful engagement and valuable feedback throughout the review process.

Sincerely,

Authors of Submission 9986

---

### Decision · Program_Chairs · 2025-09-17

**Decision:**

Accept (poster)

**Comment:**

This paper proposes a diffusion-based adversarial attack that perturbs state inputs for RL agents. The authors show that existing defenses, which are primarily designed against Lp-norm based perturbations, are vulnerable to this new attack. The method is evaluated across multiple environments and against a variety of defense mechanisms.

The authors' rebuttal successfully addressed most technical concerns. While some reservations remain regarding the attack's realism (Reviewer vqkJ) and scalability (Reviewer 2wp2), the reviewers agree on the paper's core contribution in exposing the limitations of current defense algorithms for RL agents. Therefore, we recommend acceptance.